# *CR-Net*: SCALING PARAMETER-EFFICIENT TRAINING WITH CROSS-LAYER LOW-RANK STRUCTURE

**Boao Kong**[1,*]**, Junzhu Liang**[1,*]**, Yuxi Liu**[1,*]**, Renjia Deng**[1]**, Kun Yuan**[1,†]
[1]Peking University, Beijing, China
[*]Equal contributions    [†]Corresponding author: kunyuan@pku.edu.cn

## ABSTRACT

Low-rank architectures have become increasingly important for efficient large language model (LLM) pre-training, providing substantial reductions in both parameter complexity and memory/computational demands. Despite these advantages, current low-rank methods face three critical shortcomings: (1) compromised model performance, (2) considerable computational overhead, and (3) limited activation memory savings. To address these limitations, we propose **C**ross-layer Low-**R**ank residual **Net**work (*CR-Net*), an innovative parameter-efficient framework inspired by our discovery that inter-layer activation residuals possess low-rank properties. *CR-Net* implements this insight through a dual-path architecture that efficiently reconstructs layer activations by combining previous-layer outputs with their low-rank differences, thereby maintaining high-rank information with minimal parameters. We further develop a specialized activation recomputation strategy tailored for *CR-Net* that dramatically reduces memory requirements. Extensive pre-training experiments across model scales from 60M to 7B parameters demonstrate that *CR-Net* consistently outperforms state-of-the-art low-rank frameworks while requiring fewer computational resources and less memory. Code is available at https://github.com/KongBoao/CR-Net.

## 1 INTRODUCTION

Large language models (LLMs) have achieved remarkable success across diverse domains (Brown et al., 2020; Touvron et al., 2023; Grattafiori et al., 2024; Liu et al., 2024a), with strong empirical evidence demonstrating that scaling both model parameters and training data consistently improves model performance (Kaplan et al., 2020; Hoffmann et al., 2022; Rae et al., 2021). However, as model sizes grow from millions to billions of parameters, the computational requirements for pre-training increase exponentially in both computation power and memory consumption. This necessitates months-long training cycles on large clusters of high-performance GPUs, making the process both time-prohibitive and economically challenging. For example, scaling from GPT-3's 2.7B parameter version to its 175B counterpart increases memory requirements from 21GB to 1.4TB (a 66× increase) and computational costs from 55 to 3,640 petaflop-days (another 66× increase) (Brown et al., 2020). These challenges underscore the critical need for developing efficient architectures that systematically leverage the inherent characteristics of LLM model structures to optimize both memory utilization and computational efficiency while maintaining cost-effectiveness.

**Low-rank structures in LLM training.** The low-rank property has emerged as one of the most prominent structural characteristics in transformer-based models, attracting significant research attention due to its consistent empirical validation. Existing approaches to leveraging this property can be fundamentally classified based on two principal observations:

**(O1). Low-rank parameter.** Representative approaches such as Low-Rank Adaptation (LoRA) and its variants (Hu et al., 2022; Lialin et al., 2023; Zhang et al., 2023; Liu et al., 2025; Xia et al., 2024; Kamalakara et al., 2022; Miles et al., 2024; Dettmers et al., 2023) utilize learnable low-rank parameter matrices as memory-efficient substitutes for full-rank weight updates. This design achieves significant parameter reduction while preserving model capacity and can achieve further memory savings with quantized parameters (Dettmers et al., 2023). Furthermore, this paradigm demonstrates

strong compatibility with sparsity-inducing techniques (e.g., SLTrain (Han et al., 2024) and LOST (Li et al., 2025)) to achieve improved model performance under constrained parameter budgets.

**(O2). Low-rank gradient.** Recent research has explored the intrinsic low-rank properties of LLM gradients (Zhao et al., 2024; Chen et al., 2024a; Hao et al., 2024; Huang et al.; Robert et al., 2024; He et al., 2024; Liang et al., 2024). By projecting optimizer states into low-dimensional subspaces, these methods achieve significant memory reduction during pre-training. Extensions like RSO (Chen et al., 2025) provide theoretical convergence guarantees under low-rank gradient constraints. Advanced approaches such as Apollo (Zhu et al., 2024) demonstrate additional benefits by combining channel-wise adaptive learning with low-rank gradients, yielding both improved validation accuracy and optimized memory efficiency throughput the pre-training process.

**Limitations in existing literature.** Despite their computational and storage efficiency, current approaches utilizing low-rank LLM structures suffer from several critical limitations:

**(L1). Suboptimal performance with low-rank parameter training.** Low-rank parameterization reduces memory and computational costs at the expense of suboptimal performance (Biderman et al., 2024). Techniques such as full-rank initialization (Hu et al., 2022), update aggregation (Huh et al., 2021; Lialin et al., 2023), and non-linear operators (Liu et al., 2025) can mitigate this issue but may diminish computational benefits. Furthermore, recent studies show transformer weights typically exhibit near-full-rank properties (Aghajanyan et al., 2020; Yu & Wu, 2023; Li et al., 2024), with higher-rank weights being essential for knowledge representation (Meng et al., 2022) and sensitive to low-rank approximation (Ji et al., 2024), indicating low-rank parameterization limit LLM capacity.

**(L2). Computational bottlenecks with low-rank gradient training.** While training approaches utilizing low-rank gradient empirically achieve better performance than low-rank parameterization, the gradient compression process itself introduces computational overhead. Approaches such as GaLore (Zhao et al., 2024) and FIRA (Chen et al., 2024a) exploits *Singular Value Decomposition* (SVD) to identify effective low-rank gradient subspace, which would substantially reduce throughput during training. Other methods like GoLore (He et al., 2024) and RSO (Chen et al., 2025) propose to use random low-rank gradient subspace, which sometimes leads to suboptimal performance.

**(L3). Limitations to save activation memory.** Existing approaches reduce memory load across multiple dimensions: parameters, gradients, and optimizer states. Yet a critical yet often overlooked memory burden stems from activation storage – intermediate variables cached during forward propagation required for gradient computation. Empirical studies (Zhao et al., 2024; Han et al., 2024) reveal that activation memory overhead typically ranges from 1× to 4× the model parameter size, exhibiting strong dependence on batch size configurations. While low-rank methods such as RSO (Chen et al., 2025) and CoLA-M (Liu et al., 2025) have been proposed to reduce activation memory, it remains an open question whether activations still possess potential for further compression.

**Main results.** Our contributions are threefold, forming a principled framework rather than an ad-hoc combination:

**(C1). Novel Foundational Principle:** We propose a novel principle in LLMs: the **difference** between activations of adjacent layers exhibits a strong low-rank structure. This phenomenon has been consistently observed across various models and at different training stages. Unlike low-rank properties in gradient updates or parameter modifications—such as those exploited by GaLoRE and LoRA—this structural property of activations represents an previously unreported characteristic of LLMs. It thereby serves as a **foundational basis** for more efficient pre-training paradigms.

**(C2). Parameter-efficient framework:** We propose the **C**ross-layer low-**R**ank residual **Net**work (*CR-Net*), a pre-training framework that inherently utilizes cross-layer low-rank activation differences. By computing each linear layer's activation through a combination of low-rank outputs and preceding layer activations, our design directly applies the observed low-rank structure while avoiding information loss from repeated low-rank approximations in LoRA-based approaches. This addresses **Limitations (L1)** and **(L2)** through reduced parameter complexity and memory/computation costs.

**(C3). Activation-efficient Re-computation:** we develop a re-computation strategy for *CR-Net* that removes activation storage for most layers during backward propagation. Our analysis shows superior memory efficiency and lower re-computation overhead compared to existing methods like vanilla gradient checkpointing (GCP) and CoLA-M, resolving **Limitation (L3)**.

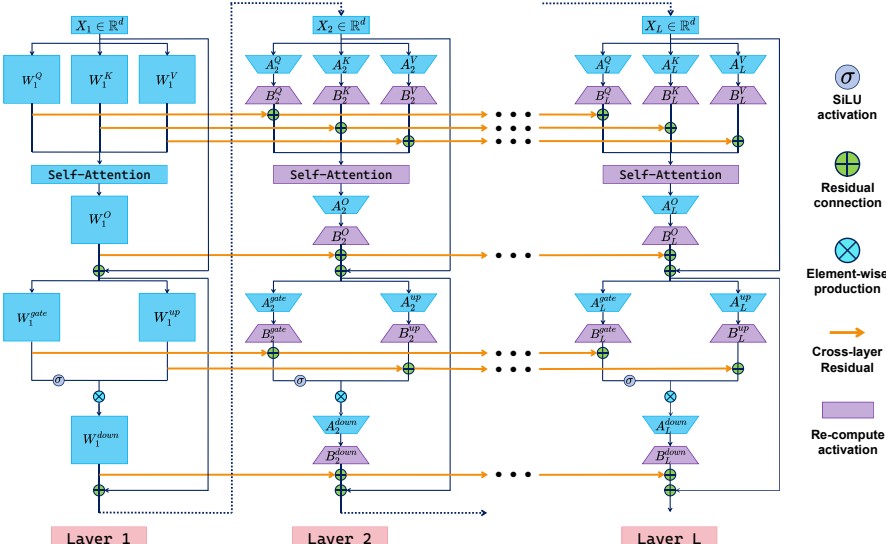

Figure 1: Illustration of *CR-Net* base on LLaMA-2 architecture with $L$ transformer layers. Layer normalization and RoPE are omitted for simplicity.

**(C4). Empirical Validation:** Large-scale pre-training experiments demonstrate that *CR-Net* achieves better validation performance than existing low-rank parameter methods while maintaining training throughput. Integration with our re-computation strategy further reduces memory consumption without compromising model capability.

## 2 RELATED WORKS AND PRELIMINARY

### 2.1 RELATED WORKS

Here we display some prior works of parameter-efficient training for LLMs. More related works can be found in Appendix A.

**Parameter-efficient training for LLMs.** The pursuit of reducing memory and computational overhead in LLM pre-training has driven significant progress in parameter-efficient training frameworks. Central to these efforts is Low-Rank Adaptation (LoRA) (Hu et al., 2022), which constrains weight updates to low-rank subspaces through matrix decomposition. This approach fundamentally differs from sparse training methodologies (Houlsby et al., 2019; Thangarasa et al., 2023) that selectively update subsets of model parameters. Recent hybrid architectures like SLTrain attempt to synergize low-rank and sparse parameterization, while multi-LoRA update strategies (Lialin et al., 2023; Xia et al., 2024) aim to recover full-rank expressiveness through sequential low-rank adjustments. Although introducing nonlinear operations to low-rank activations (Liu et al., 2025) theoretically enhances model capacity, such innovations often incur prohibitive computational costs that undermine their practicality for large-scale pre-training scenarios.

### 2.2 PRELIMINARY

**Notations.** In this paper, we assume that the amount of transformer blocks of the model is $L$. Denote $s$ and $h$ as the sequence length and hidden dimension, respectively. $h_{\text{ff}}$ stands for the intermediate dimension of the model. We use $X_l \in \mathbb{R}^{s \times h}$ to represent the input of layer $l$. $W_l^{\text{P}}$, $X_l^{\text{P}}$ and $Y_l^{\text{P}}$ stand for the parameters, inputs and outputs of linear layers at the position P. For example, $\text{P} \in \{\text{Q}, \text{K}, \text{V}, \text{O}, \text{gate}, \text{up}, \text{down}\}$ in LLaMA-based model and $\text{P} \in \{\text{Q}, \text{K}, \text{V}, \text{O}, \text{up}, \text{down}\}$ in GPT-based model. Finally, we use $\text{LR}_r(A)$ to denote the optimal approximation of $A$ with rank $r$

under Frobenius loss as

$$\mathrm{LR}_r(A) := \arg\min_{\Lambda} \|A - \Lambda\|_F^2 , \text{ s.t. } \mathrm{rank}(\Lambda) \le r.$$

**Transformer layers.** In this paper, we present our proposed *CR-Net* based on LLaMA architecture (Touvron et al., 2023; Grattafiori et al., 2024) with SwiGLU activation. And we will present the pre-training performance of *CR-Net* based on various architecture in Section 5.

We assume that the batch size is 1, unless indicated otherwise. We omit layer normalization and rotary position embedding (RoPE) operations for simplicity. Then the multi-head attention can be summarized as follows:

$$\mathrm{Att}_l = \mathrm{softmax}\left(\frac{(Y_l^{\mathrm{Q}})^\top Y_l^{\mathrm{K}}}{\sqrt{h}}\right) Y_l^{\mathrm{V}} \cdot W_l^{\mathrm{O}} + X_l,$$

$$\text{where } Y_l^{\mathrm{Q}} := X_l W_l^{\mathrm{Q}}, \quad Y_l^{\mathrm{K}} := X_l W_l^{\mathrm{K}}, \quad Y_l^{\mathrm{V}} := X_l W_l^{\mathrm{V}}. \tag{1}$$

For $W_l^{\mathrm{gate}}, W_l^{\mathrm{up}} \in \mathbb{R}^{h \times h_{\mathrm{ff}}}$ and $W_l^{\mathrm{down}} \in \mathbb{R}^{h_{\mathrm{ff}} \times h}$ as the weight (projection) matrices in the feed-forward network (FFN) in the layer $l$, where $d_{\mathrm{ff}}$ denotes the dimension of projection. Then the FFN operation can be summarized as:

$$X_{l+1} = \left(\mathrm{SwiGLU}(\mathrm{Att}_l \cdot W_l^{\mathrm{gate}}) \odot (\mathrm{Att}_l \cdot W_l^{\mathrm{up}})\right) W_l^{\mathrm{down}} + \mathrm{Att}_l, \tag{2}$$

where $\odot$ denotes the element-wise production.

# 3 *CR-Net*: CROSS-LAYER RESIDUAL NETWORK FOR LLM PRE-TRAINING

## 3.1 OBSERVATION: PREVIOUS-LAYER AND LOW-RANK TERM PRESENT A BETTER ACTIVATION APPROXIMATION

Existing studies have demonstrated that the activations in LLMs exhibit intrinsic low-rank structural properties (Cui et al., 2020; Huh et al., 2021; Yu & Wu, 2023; Liu et al., 2025) and thus low-rank approximations of activation matrices achieve acceptable reconstruction fidelity, making pre-training a model with low-rank parameters possible. Furthermore, empirical evidence from (Liu et al., 2024b; Brandon et al., 2024) highlights inter-layer activation correlations, suggesting that historical activations from preceding layers may encode critical information for current layer computations. Unlike existing observations, we have found a critical structural characteristic that **the difference between activations of adjacent layers** exhibits a significant **low-rank property**.

**Estimating activation from the historical layer and low-rank difference.** For $l = 2, 3, \cdots, L$, consider the following estimation for the activation $Y_l^{\mathrm{P}}$ as follows:

$$\widetilde{Y}_{l,\beta_0}^{\mathrm{P}} := \beta_0 Y_{l-1}^{\mathrm{P}} + \mathrm{LR}_r(\Delta_{\beta_0} Y_l^{\mathrm{P}}), \quad \text{where } \Delta_{\beta_0} Y_l^{\mathrm{P}} := Y_l^{\mathrm{P}} - \beta_0 Y_{l-1}^{\mathrm{P}}. \tag{3}$$

Here, $\beta_0$ is a tuned scaling coefficient. We aim to validate that this hybrid approach achieves superior approximation accuracy compared to direct low-rank approximation $\widetilde{Y}_{l,\mathrm{lr}}^{\mathrm{P}} = \mathrm{LR}_r(Y_l^{\mathrm{P}})$.

**Emprical evaluation for different activation approximation approaches.** To compare the performance for different methods including Eq. (3) and direct low-rank approximation for the activation estimation, we conducted comparative experiments by fine-tuning the pre-trained LLaMA-3 8B (Grattafiori et al., 2024), GPT-2-small (Radford et al., 2019), and models using the TinyShakespeare dataset. In both schemes we use the same low-rank dimension $r = 0.25h$ and we quantify approximation quality via relative error defined as follow.

$$\mathrm{Relative\ error}(Y_l^{\mathrm{P}}, \widetilde{Y}_l^{\mathrm{P}}) := \left\|\widetilde{Y}_l^{\mathrm{P}} - Y_l^{\mathrm{P}}\right\|_{\mathrm{F}} \Big/ \left\|Y_l^{\mathrm{P}}\right\|_{\mathrm{F}}, \tag{4}$$

where $\tilde{Y}_l^{\mathrm{P}}$ is an estimation of $Y_l^{\mathrm{P}}$.

Figure 2 illustrates the average relative error of two approaches for activation approximation over all the transformer layers. It can be observed that the previous-layer-based yields smaller relative errors, indicating that performing low-rank approximation with the history information provides

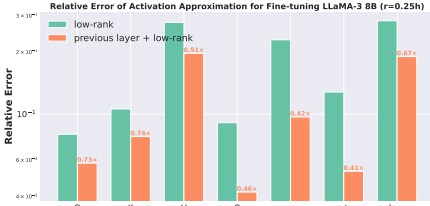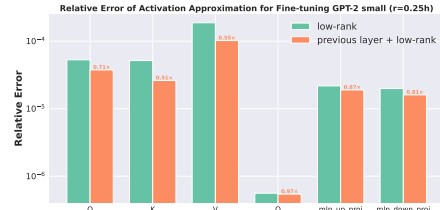

Figure 2: The average relative error of activation recovery by using low-rank approximation and using (3) over all transformer layers. (Left: LLaMA-3 8B, right: GPT-2 small.)

better reconstruction capability compared to direct approximation of the original activations when preserving the same rank $r$ of singular values. Additional evaluation can be fined in Appendix E.1. We also present a theoretical insight for this observation, whose detail is in Appendix D.

Such an observation forms the **foundational insight** of *CR-Net*, as it enables more accurate activation reconstruction with fewer parameters compared to direct low-rank approximation of $Y_l^P$, making *CR-Net* not a simple extension of existing low-rank frameworks for pre-training.

## 3.2   MODEL STRUCTURE

We utilize the empirical observation of low-rank structure for cross-layer activation differences to guide the architectural design of *CR-Net* with low-rank parameters. As evidenced by (3), the matrix $\Delta_{\beta_0} Y_l^P$ possesses an intrinsic low-rank property, which motivates the approximation as $Y_l^P \approx \beta_0 Y_{l-1}^P + \mathrm{LR}_r(\Delta_{\beta_0} Y_l^P)$.

Motivated by the low-rank cross-layer activation difference, we replace the full-shape weight $W_l^P \in \mathbb{R}^{h_{\text{in}} \times h_{\text{out}}}$ to two low-rank learnable parameter matrices $A_l^P \in \mathbb{R}^{h_{\text{in}} \times r}$ and $B_l^P \in \mathbb{R}^{r \times h_{\text{out}}}$ for $l = 2, 3, \cdots, L$, where $r < \min\{h_{\text{in}}, h_{\text{out}}\}$ is a hyper-parameter. Then the activation of layer $l$ at location P can be computed by

$$Y_l^P = \beta_0 Y_{l-1}^P + X_l^P A_l^P B_l^P. \tag{5}$$

Furthermore, to enhance the scalability of the cross-layer residual operation, we let the scaling factor $\beta_0$ as a **learnable parameter** $\beta_l^P$. With the learnable scaling factor, it can dynamically adjust the impact of the historical activation and low-rank outputs in the current activation, allowing the model to balance how much the historical information and incremental information should be carried. Specifically, when $\beta_l^P$ is near zero, the model relies heavily on the low-rank residual. Otherwise, the residual becomes a refinement over a strong propagated signal. This spectrum allows *CR-Net* to smoothly interpolate between shallow, expressive layers and deeper low-rank transitions, all within a fixed memory and computation budget. Ablations in Section 5.2 also illustrates that the learnable scaling factor can improve the training performance of *CR-Net*.

Moreover, we still use the full-size parameter matrix $W_1^P$ in the first layer. We add a normalization term $\varepsilon$ to avoid $\beta_l^P$ equals to zero, where $\varepsilon$ can be set to $10^{-6}$ in practical. At this stage, *CR-Net* change the standard matrices production to the cross-layer residual operation as follow:

$$Y_l^P = \begin{cases} X_l^P W_l^P, & l = 1, \\ \mathrm{sign}(\beta_l^P)(|\beta_l^P| + \varepsilon) Y_{l-1}^P + X_l^P A_l^P B_l^P, & l = 2, 3, \cdots, L. \end{cases} \tag{6}$$

With the low-rank parameters and cross-layer residual, we can present the framework of *CR-Net*, which can be illustrated as Figure 1. The full-rank parameter in the first layer as well as the residual for *CR-Net* can avoid the information loss by direct training with low-rank parameters.

***CR-Net*'s Stability Mechanism.** *CR-Net* overcomes the instability of low-rank pretraining by preserving high-rank activations through a full-rank first layer and modulated residual connections. A learnable scalar $\beta$ dynamically balances the high-rank state and low-rank increments, enabling robust signal reconstruction without collapsing into low-dimensional subspaces. Unlike methods that enforce strict low-rank constraints via projections or decompositions (e.g., QR, SVD), which often cause information loss and numerical issues, *CR-Net* uses standard optimization methods without additional overhead. This design ensures stable training dynamics, similar to full-rank approaches, while achieving significant parameter and memory reductions.

---

**Algorithm 1** The back-propagation of *CR-Net* with re-computation

---

**Require:** Layer inputs $X_l$ for $l = 1, 2, \cdots, L$, low-rank output $X_l^{\mathrm{P}} A_l^{\mathrm{P}}$ for $l = 1, 2, \cdots, L$, and
   linear layer activations $Y_l^{\mathrm{P}}$ for $l \in \mathcal{A}$.
  **for** $l = L, L - 1, \cdots, 1$ **do**
     Compute activations of layer $l$ for back-propagation by $X_l$, $X_l^{\mathrm{P}} A_l^{\mathrm{P}}$ and $Y_l^{\mathrm{P}}$.
     **if** $l \neq 1$ **and** $l - 1 \notin \mathcal{A}$ **then**
       **for** P in all linear layers **do**
         Reconstruct the previous-layer activation $Y_{l-1}^{\mathrm{P}}$ by Eq. (7).
       **end for**
     **end if**
  **end for**

---

### 3.3 *CR-Net* WITH ACTIVATION-EFFICIENT RE-COMPUATION

Although *CR-Net* directly reduces memory and computational costs through its parameter-efficient framework, activation values during forward propagation still consume significant GPU memory, particularly under large batch sizes.

In practice, gradient checkpointing (GCP) (Chen et al., 2016) enhances memory efficiency by storing only a subset of activations (checkpoints) during forward propagation, with the remaining activations recomputed during backpropagation. The vanilla GCP implementation requires storing layer-wise inputs while recomputing other activations through full forward passes. However, *CR-Net* introduces a critical architectural dependency: the computation of linear layer activations relies on both current low-rank outputs and historical activations from preceding layers. This dependency necessitates full forward passes through all antecedent layers during GCP recomputation, resulting in an $\mathcal{O}(L^2)$ computation overhead.

To address this, we propose a tailored recomputation strategy for *CR-Net*. The core technique of the strategy is to store a subset of linear activations. Specifically, we select a subset of Transformer layers indexed by $\mathcal{A}$ with $L \in \mathcal{A}$ and $1 \notin \mathcal{A}$ typically. During forward propagation, we store:

- All layer inputs $X_l$.
- All linear layer activations $Y_l^{\mathrm{P}}$ for layers $l \in \mathcal{A}$.
- Low-rank outputs $X_l^{\mathrm{P}} A_l^{\mathrm{P}}$ for $l = 2, 3, \cdots, L$ to minimize matrix recomputation costs.

During backpropagation (for layers $l = 1, 2, \cdots, L$), we recompute activations using stored inputs and checkpoints. If $Y_l^{\mathrm{P}}$ is stored, we can use it to obtain other activations in this transformer layer. Otherwise, it can be recovered via the inverse of the cross-layer residual connection in (6):

$$Y_l^{\mathrm{P}} = \frac{1}{\mathrm{sign}(\beta_{l+1}^{\mathrm{P}})(|\beta_{l+1}^{\mathrm{P}}| + \varepsilon)}(Y_{l+1}^{\mathrm{P}} - X_{l+1}^{\mathrm{P}} A_{l+1}^{\mathrm{P}} B_{l+1}^{\mathrm{P}}). \tag{7}$$

The back-propagation with re-computation of *CR-Net* is summarized as Algorithm 1. It enables full activation recovery using only a stored subset of activation. We also emphasize that such a storage can effectively reduce the accumulation of prediction error during the activation recovery process. By setting $|\mathcal{A}|$ as $L/8$ and storing low-rank outputs across multiple layers, such errors remain controllable (see empirical validation in Section 5.2).

## 4 COMPLEXITY ANALYSIS

In this section, we present the analysis for the parameter complexity, memory complexity, and computation complexity of *CR-Net* based on LLaMA architecture (Touvron et al., 2023; Grattafiori et al., 2024) and Adam optimizer (Kingma & Ba, 2014). We also present the **communication overhead** and **HBM memory** analysis for *CR-Net* under multiple GPU training in Appendix G.

### 4.1 COMPLEXITY ANALYSIS WITHOUT RE-COMPUTATION

**Parameter and Memory Complexity.** For *CR-Net*, the parameter complexity of the first layer is identical to that of the full-rank model. In subsequent layers, *CR-Net* employs two low-rank matrices

Table 1: Computation complexity of different efficient pre-training approaches for one gradient step based on LLaMA architecture. The optimizer are all standard Adam. Lower-order terms are omitted for brevity. The computation complexity of other methods are referred from (Liu et al., 2025).

| Approach | FLOPs |
|---|---|
| Full-rank | $L(24sh^2 + 12s^2h + 18shh_{\text{ff}})$ |
| (Re)LoRA | $L(40sh^2 + 24s^2h + 30shh_{\text{ff}})$ |
| SLTrain | $L(24sh^2 + 12s^2h + 18shh_{\text{ff}} + 24h^2r + 18hh_{\text{ff}}r)$ |
| GaLore | $L(24sh^2 + 12s^2h + 18shh_{\text{ff}} + 16h^2r + 12hh_{\text{ff}}r)$ |
| CoLA | $L(48shr + 12s^2h + 18sr(h + h_{\text{ff}}))$ |
| *CR-Net* | $24sh^2 + 12s^2h + 18shh_{\text{ff}} + (L-1)(48shr + 12s^2h + 18sr(h + h_{\text{ff}}))$ |

with $hr$ parameters to replace the full-rank matrices in self-attention, and uses two additional low-rank matrices with $hr$ and $h_{\text{ff}}r$ parameters respectively for the FFN operators. The overall parameter complexity is summarized as follows:

$$\underbrace{4h^2 + 3hh_{\text{ff}}}_{\text{Parameters in the first layer}} + \underbrace{(L-1)(11hr + 3h_{\text{ff}}r)}_{\text{Parameters in the other layers}}. \tag{8}$$

Regarding memory overhead for storing parameters, gradients, and optimizer states with the Adam optimizer, the requirement is approximately 4 times the parameter count. Consequently, the memory complexity is:

$$\underbrace{16h^2 + 12hh_{\text{ff}}}_{\text{Memory in the first layer}} + \underbrace{(L-1)(44hr + 12h_{\text{ff}}r)}_{\text{Memory in the other layers}}. \tag{9}$$

With low-rank parameters, the parameter complexity is reduced, particularly if $r \leq h/2$. In practice, setting $r \approx 0.25h$ achieves approximately 50% saving in parameters while maintaining comparable pre-training performance to full-rank training, as demonstrated in Section 5.1.

**Computational Complexity.** For the first transformer layer of *CR-Net*, the forward- and backward-propagation process maintains identical computational complexity to full-rank training. In subsequent layers, *CR-Net* employs low-rank weights while eliminating the nonlinear activation step applied to low-rank outputs, combined with cross-layer residual connections in contrast to CoLA (Liu et al., 2025). Given that both operations introduce lower-order computational complexity terms, we conclude that these layers maintain equivalent FLOPs to CoLA. Following existing analysis (Liu et al., 2025), the total computation complexity is (see Appendix B.2 for details):

$$\underbrace{24sh^2 + 12s^2h + 18shh_{\text{ff}}}_{\text{FLOPs in the first layer}} + \underbrace{(L-1)(48shr + 12s^2h + 18sr(h + h_{\text{ff}}))}_{\text{FLOPs in the other layers}}. \tag{10}$$

Table 1 demonstrates computational requirements per gradient step. As $h_{\text{ff}} \approx 8h/3$ in LLaMA-based model with SwiGLU activation, *CR-Net* achieves complexity reduction over full-rank pre-training when $r < 0.5d$. Notably, while *CR-Net* exhibits marginally higher FLOPs than CoLA with the same $r$ due to the full-size first layer, it enables superior training performance at lower $r$ values with reduced computation. See Section 5.1 for empirical validation.

## 4.2 COMPLEXITY ANALYSIS WITH RE-COMPUTATION

We evaluate recomputation overhead and activation memory costs for *CR-Net* based on LLaMA-like architecture in C. Table 2 compares activation memory and recomputation complexity across frameworks. including practical measurements for LLaMA2-7B (batch size 16). Although our re-computation strategy introduce additional memory overhead than Vanilla GCP and CoLA-M, *CR-Net* also achieves a 67.4% reduction in total computation overhead compared to Vanilla GCP and a 8.0% reduction compared to CoLA with significant memory saves, validating the effectiveness of our proposed cross-layer framework and re-computation strategy.

## 5 EXPERIMENTS

This section conducts numerical experiments to systematically assess the efficiency of the proposed *CR-Net* methodology. Our evaluation framework encompasses pre-training tasks spanning diverse

Table 2: Activation memory and re-computation complexity of different activation-efficient approaches based on LLaMA architecture with batch size 1 and BF16 precision. For *CR-Net*, the notation $b$ represents the elements of $\mathcal{A}$ where $b = 4$. The last two column is the evaluated computation and memory of LLaMA2-7B framework with $r = 512$, batch size $B = 16$, and sequence length $s = 256$. We also add a explanation of the re-computation complexity of CoLA-M in Appendix C.2.

| Algorithms | Activation Memory | Re-compute | Memory (GB) | FLOPs ($\times 10^{15}$) |
|---|---|---|---|---|
| Vamilla GCP | $Lsh$ | $24Lsh^2 + 4Ls^2h$ | 51.22 (1.000×) | 2.119 (1.000×) |
| CoLA-M | $2Lsh + 7Lsr$ | $20.67Lshr + 4Ls^2h$ | 24.78 (0.484×) | 0.752 (0.355×) |
| *CR-Net* | $(L+10b)sh + 7(L-1)sr$ | $20.67(L-b)shr + 4Ls^2h$ | 23.35 (0.456×) | 0.692 (0.326×) |
| Full-rank | $20.67Lsh + 2Ls^2a$ | N.A. | 70.97 (1.386×) | 1.608 (0.759×) |

Table 3: Comparison of validation perplexity (PPL) (↓), parameter complexity (M) (↓) and evaluated memory overhead (GB) (↓) of parameter, gradients, and optimizer states of different effective training approaches in the pre-training task of LLaMA model with C4-en dataset. The results of compared methods are referred from (Huang et al.; Han et al., 2024; Liu et al., 2025; Chen et al., 2025; Zhu et al., 2024; Miles et al., 2024; Mo et al., 2025). For *CR-Net*, it is compared with other parameter-efficient methods with aligned parameter complexity (marked as ◇) and compared with other optimizer-effiencnt methods with aligned memory overhead (marked as †). N.A. means that the corresponding experiment has not been taken.

| Approach | 60M | | | 130M | | | 350M | | | 1B | | |
|---|---|---|---|---|---|---|---|---|---|---|---|---|
| | 1.1B tokens | | | 2.2B tokens | | | 6.4B tokens | | | 13.1B tokens | | |
| | PPL | Para | Mem | PPL | Para | Mem | PPL | Para | Mem | PPL | Para | Mem |
| Full-rank | 34.06 | 58 | 0.43 | 24.36 | 134 | 1.00 | **18.80** | 368 | 2.74 | 15.56 | 1339 | 9.98 |
| LoRA | 34.99 | 58 | 0.37 | 33.92 | 134 | 0.86 | 25.58 | 368 | 1.94 | 19.21 | 1339 | 6.79 |
| ReLoRA | 37.04 | 58 | 0.37 | 29.37 | 134 | 0.86 | 29.08 | 368 | 1.94 | 18.33 | 1339 | 6.79 |
| VeLoRA | 34.35 | 58 | 0.37 | 25.88 | 134 | 0.86 | | N.A. | | | N.A. | |
| FLoRA | 33.76 | 58 | 0.37 | 25.29 | 134 | 0.86 | | N.A. | | | N.A. | |
| SLTrain | 34.15 | 44 | 0.32 | 26.04 | 97 | 0.72 | 19.42 | 194 | 1.45 | 16.14 | 646 | 4.81 |
| CoLA | 34.04 | 43 | 0.32 | 24.48 | 94 | 0.70 | 19.40 | 185 | 1.38 | 15.52 | 609 | 4.54 |
| LORO | 33.96 | 43 | 0.32 | 24.59 | 94 | 0.70 | 18.84 | 185 | 1.38 | **15.19** | 609 | 4.54 |
| *CR-Net* ◇ | **32.76** | 43 | 0.32 | **24.31** | 90 | 0.67 | 18.95 | 183 | 1.36 | 15.22 | 583 | 4.35 |
| GaLore | 34.88 | 58 | 0.36 | 25.36 | 134 | 0.79 | 18.95 | 368 | 1.90 | 15.64 | 1339 | 6.60 |
| RSO | 34.55 | 58 | 0.36 | 25.34 | 134 | 0.79 | 18.87 | 368 | 1.90 | 15.86 | 1339 | 6.60 |
| Apollo | **31.55** | 58 | 0.36 | **22.94** | 134 | 0.79 | **16.85** | 368 | 1.90 | 14.20 | 1339 | 6.60 |
| *CR-Net* † | 32.76 | 43 | 0.32 | 23.74 | 106 | 0.79 | 17.08 | 250 | 1.86 | **14.05** | 870 | 6.48 |

model scales. Furthermore, we empirically validate the performance advantages of our innovative re-computation approach. To substantiate design choices, comprehensive ablation studies are performed focusing on two critical components: the determination of optimal rank values for low-rank parameterization and the adaptive learning dynamics of scaling coefficients $\beta_l^{\mathrm{P}}$ within our parameterization scheme. More experimental results can be referred in Appendix E.

## 5.1 PRETRAINING WITH *CR-Net*

**Experiment setup.** We evaluate the proposed *CR-Net* framework by pre-training LLaMA-2 models (Touvron et al., 2023) with size varying from 60M to 13B. Following the existing experimental settings (Zhao et al., 2024), we train the model over C4-en dataset (Raffel et al., 2020), which is primarily intended for pre-training language models and word representations with large scale. We compared *CR-Net* with existing parameter-efficient methods including LoRA (Hu et al., 2022), ReLoRA (Lialin et al., 2023), VeLoRA (Miles et al., 2024), FLoRA (Hao et al., 2024), SLTrain (Han et al., 2024), SLTrain (Han et al., 2024), CoLA (Liu et al., 2025), LORO (Mo et al., 2025) **with aligned parameter complexity**. We also compared *CR-Net* with other optimizer-efficient algorithms including GaLore (Zhao et al., 2024), RSO (Chen et al., 2025), and Apollo (Zhu et al., 2024) **with aligned memory overhead**. We use the same configurations as those reported in (Zhao et al., 2024),

Table 4: Comparison of validation perplexity (↓) and memory (↓) of different approaches in LLaMA-2 7B pre-training tasks. The results of compared methods are referred from (Liu et al., 2025; Zhu et al., 2024).

| | Memory (GB) | Training steps | | | |
|---|---|---|---|---|---|
| | | 10K | 40K | 65K | 80K |
| 8-bit Adam | 72.59 | N.A. | 18.09 | N.A. | 15.47 |
| 8-bit GaLore | 65.16 | 26.87 | 17.94 | N.A. | 15.39 |
| Apollo | N.A. | N.A. | 17.55 | N.A. | 14.39 |
| CoLA-M | 28.82 | **22.76** | 16.21 | 14.59 | 13.82 |
| *CR-Net w.* re-computation | 27.60 | 23.11 | **16.01** | **14.47** | **13.72** |
| Training tokens (B) | | 1.3 | 5.2 | 8.5 | 10.5 |

Table 5: Comparison of validation perplexity (↓) of different approaches in LLaMA-2 13B pre-training tasks.

| Steps | 40K |
|---|---|
| 8-bit Adam | 17.85 |
| *CR-Net w.* re-computation | 18.12 |

Table 6: Comparison of validation perplexity (↓) for *CR-Net* with or without low-rank first layer in LLaMA-2 350M pre-training tasks.

| Training tokens | 6.4B |
|---|---|
| *CR-Net w.o.* low-rank first layer | 18.95 |
| *CR-Net w.* low-rank first layer | 19.68 |

and the detailed experiment setting are in Appendix E.2.1. For models with sizes ranging from 60M to 1B parameters, we do not employ the re-computation strategy to maintain consistency with existing baselines. For the 7B and 13B models, we employ re-computation to reduce memory consumption.

**Pre-training results.** Table 3 has shown that *CR-Net* generally outperforms other parameter-efficient with the same of lower training parameters. It can be observed that *CR-Net* even achieve a better performance as full-rank training while reducing the parameter complexity by 56.5% and the per-step computation complexity by 63.2% for LLaMA-2 1B network. When aligning the memory overhead, *CR-Net* achieves a better validation perplexity than that of optimizer-efficient algorithms especially with the model size larger than 1B, showing the benefits of *CR-Net* with large-scale scenarios.

Table 4 illustrates the validation perplexity of *CR-Net* when training with LLaMA2-7B as well as other competitive approaches. It can be observed that *CR-Net* outperforms all baselines in the training process with a lower memory overhead when using re-computation.

Furthermore, we report the validation perplexity for the pre-training of LLaMA-2 13B after 40,000 initialization steps. Experimental details are provided in Appendix E.2.3. As demonstrated in Table 5, *CR-Net* achieves more than 50% reduction in parameters, while incurring only a 2% degradation in validation performance.

**Pre-training and inference throughput.** Figure 3 illustrates the throughput performance of our method alongside comparable approaches in both pre-training and inference tasks, with detailed experimental configurations provided in Appendix E.2.1. As shown, *CR-Net* achieves superior throughput compared to other methods in LLaMA-2 1B pre-training and inference scenarios. Notably, even when accounting for data-parallel communication overhead across 4 GPUs for LLaMA-2 1B pre-training, *CR-Net* demonstrates over 6% improvement relative to state-of-the-art methods that do not incur such communication costs. Furthermore, *CR-Net* exhibits an 8.6% throughput enhancement compared to LORO in LLaMA-2 7B pre-training.

## 5.2 ABLATIONS

**How does rank selection impact pre-training performance?** We train the LLaMA-2 350B network with *CR-Net* using identical parameters but varying ranks across transformer layers. Experimental details are provided in Appendix E.2.6. Figure 4 shows that networks with higher ranks in middle layers and lower ranks in side layers exhibit superior performance.

**Whether does the learnable scaling factor $\beta_l^{\mathrm{P}}$ benefit the model convergence?** We train the LLaMA-2 350M network with the same hyperparameters as that in Section 5.1 except make the

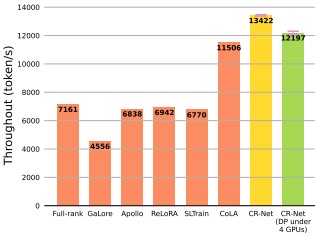 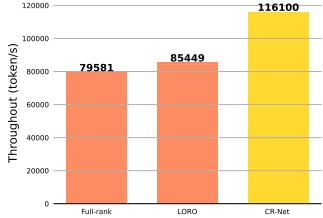 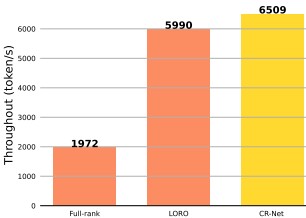

Figure 3: The average throughput (tokens/s) for each device of different algorithms. (Left: LLaMA-2 1B pre-training on an Nvidia A100 40G GPU, with results of other comparable methods from (Liu et al., 2025). Middle: LLaMA-2 1B inference on a Nvidia A100 80G GPU. Right: LLaMA-2 7B pre-training on a Nvidia A100 80G GPU.)

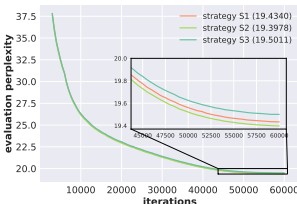 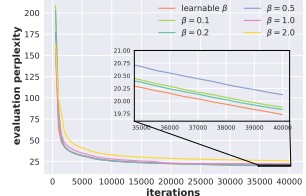 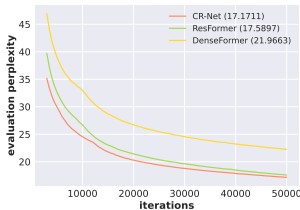

Figure 4: The evaluation perplexity for *CR-Net* in training LLaMA-2 350M model with different strategies of rank selection.

Figure 5: The Comparison of evaluation perplexity for *CR-Net* in training LLaMA-2 350M with fixed $\beta_l^{\mathrm{P}}$ and learnable $\beta_l^{\mathrm{P}}$.

Figure 6: The evaluation perplexity for *CR-Net*, Learnable-ResFormer and DenseFormer in the LLaMA-2 1B pre-training task.

scaling $\beta_l^{\mathrm{P}}$. From the perplexity shown in Figure 5, we can obtain that training *CR-Net* with a learnable $\beta_l^{\mathrm{P}}$ can improve the numerical stability as there are spikes in the runs with $\beta_l^{\mathrm{P}}$ fixed. This match our observation in Section that the best $\beta_l^{\mathrm{P}}$ to achieve the lowest rank varies during the training process. More discussions can be found in Appendix F.2.

**Whether other cross-layer residual strategies do well in pre-training with low-rank parameters?** To compare different cross-layer residual patterns under the same global low-rank parameterization, we implement a Learnable-ResFormer variant that replaces *CR-Net*'s previous-layer residual term with a learnable skip from the first layer's activation across all low-rank linear layers, inspired by the first-layer value residual in ResFormer (Zhou et al., 2024) and DenseFormer (Pagliardini et al., 2024). As Figure 6 illustrates, *CR-Net* outperforms the other two low-rank adaptation under the similar low-rank parameter budget.

**Whether the full-rank first layer is necessary?** We trained the LLaMA-2 350M model with either full-rank or low-rank parameters in the first transformer layer. As shown in Table 6, utilizing a low-rank first layer introduces a 3.5% degradation in validation perplexity. This finding demonstrates the critical role of the full-rank first layer in reconstructing high-rank signals from low-rank information.

## 6 CONCLUSION

We propose *CR-Net*, a parameter-efficient LLM pretraining framework that computes linear activations via low-rank transformations with cross-layer residuals. This dual-path design preserves high-rank capacity using fewer parameters, achieving better validation performance with reduced computation and memory than conventional low-rank frameworks.

We outline two directions for future work on *CR-Net* framework. First, system-level implementations integrating mixed-precision training could alleviate the growing memory overhead encountered when scaling *CR-Net* to larger models, particularly for activation recomputation strategies. Second, generalizing the framework's current multi-head attention foundation to alternative attention architectures, including multi-head latent attention (Liu et al., 2024a), would broaden its architectural versatility.

ACKNOWLEDGMENTS

This work is supported by the National Key Research and Development Program of China (No. 2024YFA1012902) and National Natural Science Foundation of China (No. 12288101, 92370121, 12301392). This work is also supported by AI for Science Institute, Beijing, China.

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

# Appendix

## A  ADDITIONAL RELATED WORKS

**Low-rank training for LLMs.** Beyond explicit parameter sparsity, low-rank principles permeate various aspects of optimization mechanics. The Adafactor algorithm (Shazeer & Stern, 2018) pioneers this direction by replacing full-rank momentum buffers with factorized second-moment estimators, establishing a blueprint for memory-efficient optimizers. Subsequent developments like Sophia (Liu et al., 2023) integrate low-rank Hessian approximations with lightweight curvature estimation, achieving both speed and memory advantages. The Lion optimizer (Chen et al., 2023) takes a radical approach through sign-based gradient compression, creating natural compatibility with low-rank parameterization. A paradigm shift occurs with GaLore (Zhao et al., 2024), which projects entire optimization trajectories into low-rank subspaces via Singular Value Decomposition (SVD). This framework inspires derivative works (Hao et al., 2024; He et al., 2024; Chen et al., 2025) that replace computationally intensive SVD with stochastic orthogonal projections. To address information loss inherent in low-rank approximations, error-feedback mechanisms in (Robert et al., 2024; Chen et al., 2024a) iteratively compensate for residual gradients. Meanwhile, Apollo (Zhu et al., 2024) reimagines adaptive learning rates by incorporating projected optimizer states as scaling factors, demonstrating how traditional algorithms like Adam (Kingma & Ba, 2014) can be retrofitted for low-rank efficiency.

**Activation-efficient training for LLMs.** Activation memory optimization operates through dual pathways: algorithmic innovation and system-level engineering. Zero-order optimization methods (Malladi et al., 2023; Gautam et al., 2024; Chen et al., 2024b) circumvent backpropagation by estimating gradients through forward pass perturbations, though their adoption is hindered by fundamental convergence limitations well-documented in optimization theory (Duchi et al., 2015; Berahas et al., 2022). The randomized subspace optimization (RSO) framework (Chen et al., 2025) offers a middle ground by performing gradient updates in dimension-reduced spaces, thereby implicitly reducing activation storage requirements. On the systemic front, gradient checkpointing (GCP) techniques (Chen et al., 2016; Feng & Huang, 2021; He & Yu, 2023) strategically store partial activations during forward propagation and recompute missing values during backward passes, achieving linear memory savings at the cost of increased computation. FlashAttention (Dao et al., 2022; Dao, 2023; Shah et al., 2024) revolutionizes attention layer implementations through block-wise computation and dynamic memory management, effectively decoupling peak memory demand from sequence length. These complementary approaches collectively address the "memory wall" challenge in modern LLM training.

**Cross-layer structure in transformer.** Recent research increasingly highlights distinctive data distribution patterns across transformer layers, particularly in large-scale model optimization. For inference acceleration, (Liu et al., 2024b) first identified high inter-layer KV-cache similarity, proposing shared caching mechanisms between adjacent layers–an insight further developed by (Brandon et al., 2024) through cross-layer attention operators for dynamic KV compression. In pre-training contexts, architectural innovations like token-level attention initialization (Zhou et al., 2024; Pagliardini et al., 2024; Nguyen et al., 2023) enhance information propagation across layers. However, these innovations predominantly target attention layer optimizations while inadequately addressing two persistent gaps: the interplay of feed-forward network dynamics across layers remains underexplored, and existing frameworks demonstrate limited compatibility with low-rank adaptation paradigms, constraining their applicability to parameter-efficient training scenarios.

## B  COMPUTATION AND MEMORY ANALYSIS OF *CR-Net* WITHOUT RE-COMPUTATION

In this section, we present a detailed computation and memory analysis of *CR-Net*.

### B.1 TRANSFORMER LAYERS WITH LLaMA ARCHITECTURE

In order to make a Comparison with other efficient methods, the computation and memory analysis are based on LLaMA framework (Touvron et al., 2023) with $L$ transformer layers. Here, we focus on the forward- and backward-propagation of *CR-Net*. It should be reminded that we omit the layer normalization, rotary position embedding, and in-layer residual for simply.

**Forward Propagation.** For the input $X_l \in \mathbb{R}^{s \times h}$, where $s$ denotes the sequence length and $h$ denotes the hidden dimension. For the first layer, the transformer block's attention mechanism implements a series of linear transformations through three fundamental computational stages:

$$Y_1^{\mathrm{Q}} = X_1 W_1^{\mathrm{Q}}, \quad Y_1^{\mathrm{K}} = X_1 W_1^{\mathrm{K}}, \quad Y_1^{\mathrm{V}} = X_1 W_1^{\mathrm{V}}, \tag{11}$$

where $W_1^{\mathrm{Q}}, W_1^{\mathrm{K}}, W_1^{\mathrm{V}} \in \mathbb{R}^{h \times h}$ are full-size weight parameters. For the other layers, the weight matrices of linear transformations are replaced by low-rank weights and the cross-layer residual should also be taken:

$$Y_l^{\mathrm{Q}} = \beta_l^{\mathrm{Q}} Y_{l-1}^{\mathrm{Q}} + X_l A_l^{\mathrm{Q}} B_l^{\mathrm{Q}}, \quad Y_l^{\mathrm{K}} = \beta_l^{\mathrm{K}} Y_{l-1}^{\mathrm{K}} + X_l A_l^{\mathrm{K}} B_l^{\mathrm{K}}, \quad Y_l^{\mathrm{V}} = \beta_l^{\mathrm{V}} Y_{l-1}^{\mathrm{V}} + X_l A_l^{\mathrm{V}} B_l^{\mathrm{V}}, \tag{12}$$

where $A_l^{\mathrm{Q}}, A_l^{\mathrm{K}}, A_l^{\mathrm{V}} \in \mathbb{R}^{h \times r}$ and $B_l^{\mathrm{Q}}, B_l^{\mathrm{K}}, B_l^{\mathrm{V}} \in \mathbb{R}^{r \times h}$ are low-rank parameter matrices. Then for $l = 1, 2, \cdots, L$, the immediate activations can be combined as follows:

$$\widetilde{\mathrm{Att}}_l^{\mathrm{s}} = Y_l^{\mathrm{Q}}(Y_l^{\mathrm{K}})^{\top}, \quad \mathrm{Att}_l^{\mathrm{s}} = \mathrm{softmax}\left(\frac{\widetilde{\mathrm{Att}}_l^{\mathrm{s}}}{\sqrt{h}}\right), \quad \mathrm{Att}_l^{\mathrm{h}} = \mathrm{Att}_l^{\mathrm{s}} Y_l^{\mathrm{V}}. \tag{13}$$

Then for the first layer, the attention output can be obtained by:

$$Y_1^{\mathrm{O}} = \mathrm{Att}_1^{\mathrm{h}} W_1^{\mathrm{O}}, \tag{14}$$

where $W_1^{\mathrm{O}} \in \mathbb{R}^{h \times h}$ denotes the full-size output matrix. For $l = 2, 3, \cdots, L$, the attention output can be obtained by:

$$Y_l^{\mathrm{O}} = \beta_l^{\mathrm{O}} Y_{l-1}^{\mathrm{O}} + \mathrm{Att}_l^{\mathrm{h}} A_l^{\mathrm{O}} B_l^{\mathrm{O}}, \tag{15}$$

where $A_l^{\mathrm{O}} \in \mathbb{R}^{h \times r}$ and $B_l^{\mathrm{O}} \in \mathbb{R}^{h \times r}$ denote the low-rank parameter matrices.

Next, the feed-forward network consists of three linear layers. Among them, the gate layer and up-projection layer are computed in layer 1 as:

$$Y_1^{\mathrm{gate}} = Y_1^{\mathrm{O}} W_1^{\mathrm{gate}}, \quad Y_1^{\mathrm{up}} = Y_1^{\mathrm{O}} W_1^{\mathrm{up}}, \quad X_1^{\mathrm{down}} = \mathrm{SwiGLU}(Y_1^{\mathrm{gate}}) \odot Y_1^{\mathrm{up}}, \tag{16}$$

where $W_1^{\mathrm{gate}}, W_1^{\mathrm{up}} \in \mathbb{R}^{h \times h_{\mathrm{ff}}}$ are full-size weight matrices. For the other layer, it can be computed as:

$$\begin{aligned} Y_l^{\mathrm{gate}} &= \beta_l^{\mathrm{gate}} Y_{l-1}^{\mathrm{gate}} + Y_l^{\mathrm{O}} A_l^{\mathrm{gate}} B_l^{\mathrm{gate}}, \quad Y_l^{\mathrm{up}} = \beta_l^{\mathrm{up}} Y_{l-1}^{\mathrm{up}} + Y_l^{\mathrm{O}} A_l^{\mathrm{up}} B_l^{\mathrm{up}}, \\ X_l^{\mathrm{down}} &= \mathrm{SwiGLU}(Y_l^{\mathrm{gate}}) \odot Y_l^{\mathrm{up}}, \end{aligned} \tag{17}$$

where $A_l^{\mathrm{gate}}, A_l^{\mathrm{up}} \in \mathbb{R}^{h \times r}$ and $B_l^{\mathrm{gate}}, B_l^{\mathrm{up}} \in \mathbb{R}^{r \times h_{\mathrm{ff}}}$ denote low-rank parameters and $\odot$ denotes the element-wise production. Finally, the down-projection in the first layer can be computed as:

$$Y_1^{\mathrm{down}} = X_1^{\mathrm{down}} W_1^{\mathrm{down}}, \tag{18}$$

where $W_1^{\mathrm{down}} \in \mathbb{R}^{h_{\mathrm{ff}} \times h}$ are full-size weight matrices. For the other layer, it can be computed as:

$$Y_l^{\mathrm{down}} = \beta_l^{\mathrm{down}} Y_{l-1}^{\mathrm{down}} + X_l^{\mathrm{down}} A_l^{\mathrm{down}} B_l^{\mathrm{down}}, \tag{19}$$

where $A_1^{\mathrm{down}} \in \mathbb{R}^{h_{\mathrm{ff}} \times r}$ and $B_1^{\mathrm{down}} \in \mathbb{R}^{r \times h}$ denote low-rank parameters.

**Backward Propagation.** For a matrix $M$ and the loss function $\ell$, we use $\mathcal{D}M := \frac{\partial \ell}{\partial M}$. To compute the gradients of all the parameters in layer $l$, the back propagation begins with the partial gradient of the loss function $\ell$ with respect to the input of the next layer, i.e., $\mathcal{D}X_{l+1}$.

Then, for the last layer, the gradient of the output holds that $\mathcal{D}Y_L^{\mathrm{down}} = \mathcal{D}X_{L+1}$, where $X_{L+1}$ denotes the input of output layers. However, for the other layer, it holds that:

$$\mathcal{D}Y_l^{\mathrm{down}} = \beta_{l+1}^{\mathrm{down}} \mathcal{D}Y_{l+1}^{\mathrm{down}} + \mathcal{D}X_{L+1}. \tag{20}$$

For the first layer, the gradient of inputs and weights for down-projection can be computed as:

$$\mathcal{D}X_1^{\text{down}} = \mathcal{D}Y_1^{\text{down}}(W_1^{\text{down}})^\top, \quad \mathcal{D}W_1^{\text{down}} = (X_1^{\text{down}})^\top \mathcal{D}Y_1^{\text{down}}. \tag{21}$$

For the other layers, the gradient of inputs and weights for down-projection can be computed as:

$$\begin{aligned}
\mathcal{D}X_l^{\text{down}} &= \mathcal{D}Y_l^{\text{down}}(A_l^{\text{down}}B_l^{\text{down}})^\top, \quad \mathcal{D}B_l^{\text{down}} = (X_l^{\text{down}}A_l^{\text{down}})^\top \mathcal{D}Y_l^{\text{down}}, \\
\mathcal{D}A_l^{\text{down}} &= (X_l^{\text{down}})^\top \mathcal{D}B_l^{\text{down}}.
\end{aligned} \tag{22}$$

For the last layer, as the gradient of activations do not rely on the former layers. Thus the gradient of $Y_L^{\text{gate}}$ and $Y_L^{\text{up}}$ can be obtained as:

$$\mathcal{D}Y_L^{\text{up}} = \mathcal{D}X_L^{\text{down}} \odot \text{SwiGLU}(Y_L^{\text{gate}}), \quad \mathcal{D}Y_L^{\text{gate}} = \mathcal{D}X_L^{\text{down}} \odot Y_L^{\text{up}} \odot \text{SwiGLU}'(Y_L^{\text{gate}}). \tag{23}$$

For the other layers, the corresponding gradient of activations should combine the back-propagated gradients and the historical gradients. Thus it holds that:

$$\begin{aligned}
\mathcal{D}Y_l^{\text{up}} &= \beta_{l+1}^{\text{up}}\mathcal{D}Y_{l+1}^{\text{up}} + \mathcal{D}X_l^{\text{down}} \odot \text{SwiGLU}(Y_l^{\text{gate}}), \\
\mathcal{D}Y_l^{\text{gate}} &= \beta_{l+1}^{\text{gate}}\mathcal{D}Y_{l+1}^{\text{gate}} + \mathcal{D}X_l^{\text{down}} \odot Y_l^{\text{up}} \odot \text{SwiGLU}'(Y_l^{\text{gate}}).
\end{aligned} \tag{24}$$

Then the gradient for parameter of up-projection and gate in the first layer can be computed by:

$$\begin{aligned}
\mathcal{D}Y_1^{\text{O}} &= \beta_2^{\text{O}}\mathcal{D}Y_2^{\text{O}} + \mathcal{D}Y_1^{\text{up}}(W_1^{\text{up}}) + \mathcal{D}Y_1^{\text{gate}}(W_1^{\text{gate}}), \\
\mathcal{D}W_1^{\text{up}} &= (Y_1^{\text{O}})^\top \mathcal{D}Y_1^{\text{up}}, \quad \mathcal{D}W_1^{\text{gate}} = (Y_1^{\text{O}})^\top \mathcal{D}Y_1^{\text{gate}}.
\end{aligned} \tag{25}$$

For $l = 2, 3, \cdots, L-1$, the gradient for parameter of up-projection and gate in the layer $l$ can be computed by:

$$\begin{aligned}
\mathcal{D}Y_l^{\text{O}} &= \beta_{l+1}^{\text{O}}\mathcal{D}Y_{l+1}^{\text{O}} + \mathcal{D}Y_l^{\text{up}}(A_l^{\text{up}}B_l^{\text{up}}) + \mathcal{D}Y_l^{\text{gate}}(A_l^{\text{gate}}B_l^{\text{gate}}), \\
\mathcal{D}B_l^{\text{up}} &= (Y_l^{\text{O}}A_l^{\text{up}})^\top \mathcal{D}Y_l^{\text{up}}, \quad \mathcal{D}A_l^{\text{up}} = (Y_l^{\text{O}})^\top \mathcal{D}B_l^{\text{up}} \\
\mathcal{D}B_l^{\text{down}} &= (Y_l^{\text{O}}A_l^{\text{down}})^\top \mathcal{D}Y_l^{\text{down}}, \quad \mathcal{D}A_l^{\text{down}} = (Y_l^{\text{O}})^\top \mathcal{D}B_l^{\text{down}}.
\end{aligned} \tag{26}$$

The gradient for parameter of up-projection and gate in the last layer can be computed by:

$$\begin{aligned}
\mathcal{D}Y_L^{\text{O}} &= \mathcal{D}Y_L^{\text{up}}(A_L^{\text{up}}B_L^{\text{up}}) + \mathcal{D}Y_L^{\text{gate}}(A_L^{\text{gate}}B_L^{\text{gate}}), \\
\mathcal{D}B_L^{\text{up}} &= (Y_L^{\text{O}}A_L^{\text{up}})^\top \mathcal{D}Y_L^{\text{up}}, \quad \mathcal{D}A_L^{\text{up}} = (Y_L^{\text{O}})^\top \mathcal{D}B_L^{\text{up}} \\
\mathcal{D}B_L^{\text{down}} &= (Y_L^{\text{O}}A_L^{\text{down}})^\top \mathcal{D}Y_L^{\text{down}}, \quad \mathcal{D}A_L^{\text{down}} = (Y_L^{\text{O}})^\top \mathcal{D}B_L^{\text{down}}.
\end{aligned} \tag{27}$$

Then for the first layer, the gradients for parameter of output layer in attention can be computed by:

$$\mathcal{D}\text{Att}_1^{\text{h}} = \mathcal{D}Y_1^{\text{O}}(W_1^{\text{O}})^\top, \quad \mathcal{D}W_1^{\text{O}} = (\text{Att}_1^{\text{h}})^\top \mathcal{D}Y_1^{\text{O}}. \tag{28}$$

For the other layers, the gradients can be computed by:

$$\mathcal{D}\text{Att}_l^{\text{h}} = \mathcal{D}Y_l^{\text{O}}(A_l^{\text{O}}B_l^{\text{O}})^\top, \quad \mathcal{D}B_l^{\text{O}} = (\text{Att}_l^{\text{h}}A_l^{\text{O}})^\top \mathcal{D}Y_l^{\text{O}}, \quad \mathcal{D}A_l^{\text{O}} = (\text{Att}_l^{\text{h}})^\top \mathcal{D}B_l^{\text{O}}. \tag{29}$$

Next, the gradient for $\text{Att}_l^{\text{s}}$ can be computed by:

$$\mathcal{D}\text{Att}_l^{\text{s}} = \text{Att}_l^{\text{h}}(Y_l^{\text{V}})^\top. \tag{30}$$

Then for the last layer, the gradient for the activation $Y_L^{\text{Q}}, Y_L^{\text{K}}, Y_L^{\text{V}}$ can be obtained by:

$$\begin{aligned}
\mathcal{D}Y_L^{\text{V}} &= (\text{Att}_L^{\text{s}})^\top \mathcal{D}\text{Att}_L^{\text{h}}, \quad \mathcal{D}Y_L^{\text{Q}} = \left[ \mathcal{D}\text{Att}_L^{\text{s}} \odot \frac{1}{\sqrt{h}}\text{softmax}'\left(\frac{\widetilde{\text{Att}}_L^{\text{s}}}{\sqrt{h}}\right)\right] Y_L^{\text{K}}, \\
\mathcal{D}Y_L^{\text{K}} &= \left[ \mathcal{D}\text{Att}_L^{\text{s}} \odot \frac{1}{\sqrt{h}}\text{softmax}'\left(\frac{\widetilde{\text{Att}}_L^{\text{s}}}{\sqrt{h}}\right)\right]^\top Y_L^{\text{Q}}.
\end{aligned} \tag{31}$$

And the gradients of the other layers can be obtained by:

$$\mathcal{D}Y_l^{\mathrm{V}} = \beta_{l+1}^{\mathrm{V}} \mathcal{D}Y_{l+1}^{\mathrm{V}} + (\mathrm{Att}_l^{\mathrm{s}})^{\top} \mathcal{D}\mathrm{Att}_l^{\mathrm{h}},$$

$$\mathcal{D}Y_l^{\mathrm{Q}} = \beta_{l+1}^{\mathrm{Q}} \mathcal{D}Y_{l+1}^{\mathrm{Q}} + \left[ \mathcal{D}\mathrm{Att}_l^{\mathrm{s}} \odot \frac{1}{\sqrt{h}} \mathrm{softmax}' \left( \frac{\widetilde{\mathrm{Att}}_l^{\mathrm{s}}}{\sqrt{h}} \right) \right] Y_l^{\mathrm{K}},$$

$$\mathcal{D}Y_l^{\mathrm{K}} = \beta_{l+1}^{\mathrm{K}} \mathcal{D}Y_{l+1}^{\mathrm{K}} + \left[ \mathcal{D}\mathrm{Att}_l^{\mathrm{s}} \odot \frac{1}{\sqrt{h}} \mathrm{softmax}' \left( \frac{\widetilde{\mathrm{Att}}_l^{\mathrm{s}}}{\sqrt{h}} \right) \right]^{\top} Y_l^{\mathrm{Q}}.$$

(32)

Finally, the gradients of the weights for query, key, and value in the first layer can be obtained as:

$$\mathcal{D}W_1^{\mathrm{Q}} = (X_1)^{\top} \mathcal{D}Y_1^{\mathrm{Q}}, \quad \mathcal{D}W_1^{\mathrm{K}} = (X_1)^{\top} \mathcal{D}Y_1^{\mathrm{K}}, \quad \mathcal{D}W_1^{\mathrm{V}} = (X_1)^{\top} \mathcal{D}Y_1^{\mathrm{V}},$$

$$\mathcal{D}X_1 = \mathcal{D}Y_1^{\mathrm{Q}} (W_1^{\mathrm{Q}})^{\top} + \mathcal{D}Y_1^{\mathrm{K}} (W_1^{\mathrm{K}})^{\top} + \mathcal{D}Y_1^{\mathrm{V}} (W_1^{\mathrm{V}})^{\top}.$$

(33)

the gradients of the weights for query, key, and value in the other layers can be obtained as:

$$\mathcal{D}B_l^{\mathrm{Q}} = (X_l A_l^{\mathrm{Q}})^{\top} \mathcal{D}Y_l^{\mathrm{Q}}, \quad \mathcal{D}B_l^{\mathrm{K}} = (X_l A_l^{\mathrm{K}})^{\top} \mathcal{D}Y_l^{\mathrm{K}}, \quad \mathcal{D}B_l^{\mathrm{V}} = (X_l A_l^{\mathrm{V}})^{\top} \mathcal{D}Y_l^{\mathrm{V}},$$

$$\mathcal{D}A_l^{\mathrm{Q}} = (X_l)^{\top} \mathcal{D}B_l^{\mathrm{Q}}, \quad \mathcal{D}A_l^{\mathrm{K}} = (X_l)^{\top} \mathcal{D}B_l^{\mathrm{K}}, \quad \mathcal{D}A_l^{\mathrm{V}} = (X_l)^{\top} \mathcal{D}B_l^{\mathrm{V}},$$

$$\mathcal{D}X_l = \mathcal{D}Y_l^{\mathrm{Q}} (A_l^{\mathrm{Q}} B_l^{\mathrm{Q}})^{\top} + \mathcal{D}Y_l^{\mathrm{K}} (A_l^{\mathrm{K}} B_l^{\mathrm{K}})^{\top} + \mathcal{D}Y_l^{\mathrm{V}} (A_l^{\mathrm{V}} B_l^{\mathrm{V}})^{\top}.$$

(34)

### B.2 COMPUTATION ANALYSIS WITHOUT RE-COMPUTATION

Here we present the computation analysis of *CR-Net* with re-computation in detail. The brief analysis is shown in Section 4.1. To begin with, we should remind that the matrices production with size $m \times n$ and $n \times r$ need $2mnr$ FLOPs.

For the first layer, the computation of forward propagation is totally the same as full-rank training. Thus the computation FLOPs can be computed by:

- Attention Q, K, V: Three matrices productions with size $s \times h$ and $h \times h$, $6sh^2$ FLOPs in total.
- Attention SDP: One matrices production with size $s \times h$ and $h \times s$ and one matrices production with size $s \times s$ and $s \times h$, $4s^2h$ FLOPs in total.
- Attention O: One matrices production with size $s \times h$ and $h \times h$, $2sh^2$ FLOPs in total.
- FFN gate and up: Two matrices productions with size $s \times h$ and $h \times h_{\mathrm{ff}}$, $4shh_{\mathrm{ff}}$ FLOPs in total.
- FFN down: One matrices production with size $s \times h_{\mathrm{ff}}$ and $h_{\mathrm{ff}} \times h$, $2shh_{\mathrm{ff}}$ FLOPs in total.

Thus, the computation complexity of the first layer is

$$8sh^2 + 4s^2h + 6shh_{\mathrm{ff}}.$$

For the other layers, we can omit the cross-layer residual operations as its complexity is a lower-order term. Thus, the computation complexity of different components are as follow:

- Attention Q, K, V: Three matrices productions with size $s \times h$ and $h \times r$, and three matrices productions with size $s \times r$ and $r \times h$, $12shr$ FLOPs in total.
- Attention SDP: The same as that of the first layer, $4s^2h$ FLOPs in total.
- Attention O: One matrices production with size $s \times h$ and $h \times r$, and one matrices production with size $s \times r$ and $r \times h$, $4shr$ FLOPs in total.
- FFN gate and up: Two matrices productions with size $s \times h$ and $h \times r$, and two matrices productions with size $s \times r$ and $r \times h_{\mathrm{ff}}$, $4sr(h + h_{\mathrm{ff}})$ FLOPs in total.
- FFN down: One matrices production with size $s \times h$ and $h \times r$, and one matrices production with size $s \times r$ and $r \times h_{\mathrm{ff}}$, $2sr(h + h_{\mathrm{ff}})$ FLOPs in total.

Thus, the computation complexity of the other layers is

$$16shr + 4s^2h + 6sr(h + h_{\mathrm{ff}}).$$

Finally, the total computation complexity in the forward-propagation is:

$$\underbrace{8sh^2 + 4s^2h + 6shh_{\text{ff}}}_{\text{FLOPs in the first layer}} + (L-1)\underbrace{(16shr + 4s^2h + 6sr(h + h_{\text{ff}}))}_{\text{FLOPs in the other layers}}. \tag{35}$$

**Backward-propagation.** For backward-propagation. It can be obtained that the toatl computation FLOPs is twice of that in forward-propagation if we omit the lower-order terms. Thus, the total computation complexity in the forward-propagation is:

$$\underbrace{16sh^2 + 8s^2h + 12shh_{\text{ff}}}_{\text{FLOPs in the first layer}} + (L-1)\underbrace{(32shr + 8s^2h + 12sr(h + h_{\text{ff}}))}_{\text{FLOPs in the other layers}}. \tag{36}$$

Thus, we can obtain that the total computation of *CR-Net* for one gradient step is:

$$24sh^2 + 12s^2h + 18shh_{\text{ff}} + (L-1)(48shr + 12s^2h + 18sr(h + h_{\text{ff}})). \tag{37}$$

# C  COMPUTATION AND MEMORY ANALYSIS WITH RE-COMPUTATION

## C.1  ACTIVATION MEMORY AND RE-COMPUTATION ANALYSIS OF *CR-Net*

In this section, we present the computation and memory analysis of *CR-Net* with re-computation strategy which has been shown in Section 3.3.

### C.1.1  MEMORY ANALYSIS

We first present the analysis of memory overheads of *CR-Net* with re-computation. In fact, the following variables need to be stored:

- Inputs for each layer: $Lsh$ parameters in total.
- Low-rank outputs $X_l^{\text{P}} A_l^{\text{P}}$ for all linear layers except the first transformer layer: $7(L-1)sr$ parameters in total.
- Linear outputs $X_l^{\text{P}} A_l^{\text{P}} B_l^{\text{P}}$ for $l \in \mathcal{A}$: $5|\mathcal{A}|sh + 2|\mathcal{A}|sh_{\text{ff}}$ parameters in total.

Thus, the total memory overhead of *CR-Net* with re-computation is:

$$(L + 5|\mathcal{A}|)sh + 2|\mathcal{A}|sh_{\text{ff}} + 7(L-1)sr. \tag{38}$$

### C.1.2  COMPUTATION ANALYSIS

Then we present the analysis of re-computation overheads of *CR-Net*, which can be computed by:

- Attention Q, K, V: If $l \in \mathcal{A}$, no FLOPs need as all the activation with is necessary for the backward-propagation has been stored. If $l + 1 \notin \mathcal{A}$, the linear combination between low-rank output and the matrix $B_l^{\text{Q}}, B_l^{\text{K}}, B_l^{\text{V}}$ and the stored matrices $X_l A_l^{\text{Q}}, X_l A_l^{\text{K}}, X_l A_l^{\text{V}}$ respectively. The shape of the multiplied matrices are $s \times r$ and $r \times h$ respectively. Thus the total FLOPs is $6(L-|\mathcal{A}|)shr$.
- Attention SDP: All the layers should take one matrices production with size $s \times h$ and $h \times s$ and one matrices production with size $s \times s$ and $s \times h$. Thus the total FLOPs is $4Ls^2h$.
- Attention O: It is the same as the discussion for attention Q, K, V. Thus the total FLOPs is $2(L-|\mathcal{A}|)shr$.
- FFN gate and up: It is the same as the discussion for attention Q, K, V. Note that the size of the multiplied matrices are $s \times r$ and $rtimesh_{ff}$ respectively, thus the total FLOPs is $4(L-|\mathcal{A}|)sh_{\text{ff}}r$.
- FFN down: It is the same as the discussion for attention Q, K, V. Note that the size of the multiplied matrices are $s \times r$ and $rtimesh$ respectively, thus the total FLOPs is $2(L-|\mathcal{A}|)shr$.

Thus, the total re-computation complexity is:

$$10(L-|\mathcal{A}|)shr + 4(L-|\mathcal{A}|)sh_{\text{ff}}r + 4Ls^2h. \tag{39}$$

## C.2 RE-COMPUTATION COMPLEXITY OF COLA-M

Table 2 presents the recomputation complexity of activation-effective approaches, including CoLA-M. In fact, the re-computation complexity of CoLA-M can be obtained by the similar analysis process of *CR-Net* in Appendix C.1.2. The total re-computation complexity for one gradient step of CoLA-M is:

$$10Lshr + 4Lsh_{\text{ff}}r + 4Ls^2h. \tag{40}$$

When taking the intermediate dimension $h_{\text{ff}} = 8h/3$, the total re-computation complexity for one gradient step of CoLA-M is:

$$20.67Lshr + 4Ls^2h. \tag{41}$$

In Liu et al. (2025), the re-computation FLOPs of CoLA-M is obtained as the re-computation only takes half of total flops of linear layers with the assumption. However, for gate and up-projection layer, the production of $X_l^{\text{gate}} \in \mathbb{R}^{s \times h}$ ($X_l^{\text{up}}$) and $A_l^{\text{gate}} \in \mathbb{R}^{h \times r}$ ($A_l^{\text{up}}$) takes $2shr$ FLOPs while production of $X_l^{\text{gate}} A_l^{\text{gate}} \in \mathbb{R}^{r \times h_{\text{ff}}}$ ($X_l^{\text{up}} A_l^{\text{up}}$) and $B_l^{\text{gate}} \in \mathbb{R}^{r \times h_{\text{ff}}}$ ($B_l^{\text{up}}$) takes $2sh_{\text{ff}}r$ FLOPs. For down-projection layer, the production of $X_l^{\text{down}} \in \mathbb{R}^{s \times h_{\text{ff}}}$ and $A_l^{\text{down}} \in \mathbb{R}^{h_{\text{ff}} \times r}$ takes $2sh_{\text{ff}}r$ FLOPs while production of $X_l^{\text{down}} A_l^{\text{down}} \in \mathbb{R}^{s \times r}$ and $B_l^{\text{down}} \in \mathbb{R}^{r \times h}$ takes $2shr$ FLOPs. Thus, the computational FLOPs of the FFN are not given by $3sh_{\text{ff}}r + 3shr$—half of the total FLOPs—but rather by $4sh_{\text{ff}}r + 2shr$.

# D THEORETICAL INSIGHT: RESIDUAL SUBTRACTION IMPROVES LOW-RANK APPROXIMATION

In this section, we present a theoretical insight to explain the phenomenon we observed in Section 3.1. To begin with, we present an assumption of the cosine similarity of the activations for adjacent layers.

**Assumption 1** (Cosine similarity of adjacent activations). *For $l = 2, 3, \cdots, L$, let $Y_l^P \in \mathbb{R}^{d \times d}$ as the activation of the linear of position P for the l-th layer. There exists a constant $\varepsilon \in (0, 1)$ such that:*

$$\frac{\langle Y_l^P, Y_{l-1}^P \rangle_F}{\|Y_l^P\|_F \cdot \|Y_{l-1}^P\|_F} \geq 1 - \varepsilon,$$

*where $\langle \cdot, \cdot \rangle_F$ denotes the inner production of matrices induced by Frobenius norm.*

Assumption 1 illustrates that the activation of Transformer-based models have a high cosine similarity. Such a characteristic have been observed in different kinds of models and different positions (Liu et al., 2024b; Hao et al., 2025; Jiang et al., 2024).

Moreover, we present the definition of the stable rank.

**Definition 1** (Stable rank). *For $A \in \mathbb{R}^{d \times d}$, the stable rank of A (represented as $\varphi(A)$) is denoted as the ratio between the square of Frobenius norm of A and the square of $\ell_2$-norm of A. Specifically, it holds that*

$$\varphi(A) = \frac{\|A\|_F^2}{\|A\|_2^2}.$$

**Remark 1.** *The stable rank can be also defined as the ratio between the quadratic sum of singular values and the square of the maximum singular value.*

Then we can obtain the theorem that the approximation in Eq. (3) is a better estimator than low-rank approximation, especially with a small rank $r$.

**Theorem 1.** *Suppose Assumption 1 holds. Then there exists $r_0 > 0$ such that the approximation $\widetilde{Y}_{l,\beta}^P$ obtained by Eq. (3) has a lower error than the direct low-rank approximation $LR_r(Y_l^P)$ by a properly-selected $\beta$ if $r < r_0$. Specifically, it holds that:*

$$\left\| Y_l^P - \widetilde{Y}_{l,\beta}^P \right\|_F^2 \leq \left\| Y_l^P - LR_r(Y_l^P) \right\|_F^2.$$

*Proof.* First, we denote $\sigma_i(\cdot)$ as the $i$-th singular value of a matrix, which is listed in descending order. Then, the term $\Delta_\beta Y_l^P$ holds that:

$$\left\| \Delta_\beta Y_l^P \right\|_F^2 = \left\| Y_l^P - \beta Y_{l-1}^P \right\|_F^2 = \left\| Y_l^P \right\|_F^2 - 2\beta \langle Y_l^P, Y_{l-1}^P \rangle + \beta^2 \left\| Y_{l-1}^P \right\|_F^2. \tag{42}$$

Thus

$$\left\|Y_l^P\right\|_F^2 - \left\|\Delta_\beta Y_l^P\right\|_F^2 = 2\beta \left\langle Y_l^P, Y_{l-1}^P\right\rangle - \beta^2 \left\|Y_{l-1}^P\right\|_F^2. \tag{43}$$

From Weyl's inequality, the $i$-th singular value of $Y_l^P$ and $\Delta_\beta Y_l^P$ holds that:

$$|\sigma_i(Y_l^P) - \sigma_i(\Delta_\beta Y_l^P)| \le \left\|Y_l^P - \Delta_\beta Y_l^P\right\|_2 \le \beta \left\|Y_{l-1}^P\right\|_2. \tag{44}$$

Thus, it holds that:

$$\begin{aligned}
\sigma_i^2(Y_l^P) - \sigma_i^2(\Delta_\beta Y_l^P) =& (\sigma_i(Y_l^P) - \sigma_i(\Delta_\beta Y_l^P))(\sigma_i(Y_l^P) + \sigma_i(\Delta_\beta Y_l^P)) \\
\le& \beta \left\|Y_{l-1}^P\right\|_2 (\sigma_i(Y_l^P) + \sigma_i(\Delta_\beta Y_l^P)) \\
\le& \beta \left\|Y_{l-1}^P\right\|_2 \left(2\sigma_i(Y_l^P) + \beta \left\|Y_{l-1}^P\right\|_2\right).
\end{aligned} \tag{45}$$

Taking summation over $i = 1, 2, \cdots, r$, where $1 \le r \le d$, it holds that:

$$\begin{aligned}
\sum_{i=1}^r \left(\sigma_i^2(Y_l^P) - \sigma_i^2(\Delta_\beta Y_l^P)\right)^2 \le& \beta \left\|Y_{l-1}^P\right\|_2 \left(2\sum_{i=1}^r \sigma_i(Y_l^P) + \beta r \left\|Y_{l-1}^P\right\|_2\right) \\
\le& \beta \left\|Y_{l-1}^P\right\|_2 \left(2\sqrt{r}\left(\sum_{i=1}^r \sigma_i^2(Y_l^P)\right)^{1/2} + \beta r \left\|Y_{l-1}^P\right\|_2\right) \\
\le& \beta \left\|Y_{l-1}^P\right\|_2 \left(2\sqrt{r}\left\|Y_l^P\right\|_F + \beta r \left\|Y_{l-1}^P\right\|_2\right),
\end{aligned} \tag{46}$$

where the second inequality is due to Cauchy-Schwarz inequality.

Next, we consider the approximation error for $\mathrm{LR}_r(Y_l^P)$ and $\widetilde{Y}_{l,\beta}^P$. We can obtain that:

$$\begin{aligned}
&\left\|Y_l^P - \widetilde{Y}_{l,\beta}^P\right\|_F^2 - \left\|Y_l^P - \mathrm{LR}_r(Y_l^P)\right\|_F^2 \\
=& \left(\left\|\Delta_\beta Y_l^P\right\|_F^2 - \sum_{i=1}^r \sigma_i^2(\Delta_\beta Y_l^P)\right) - \left(\left\|Y_l^P\right\|_F^2 - \sum_{i=1}^r \sigma_i^2(Y_l^P)\right) \\
\le& -2\beta \left\langle Y_l^P, Y_{l-1}^P\right\rangle + \beta^2 \left\|Y_{l-1}^P\right\|_F^2 + 2\beta\sqrt{r}\left\|Y_{l-1}^P\right\|_2 \left\|Y_l^P\right\|_F + \beta^2 r \left\|Y_{l-1}^P\right\|_2^2 \\
\le& -2\beta(1-\varepsilon)\left\|Y_{l-1}^P\right\|_F \left\|Y_l^P\right\|_F + \beta^2 \left\|Y_{l-1}^P\right\|_F^2 + 2\beta\sqrt{r}\left\|Y_{l-1}^P\right\|_2 \left\|Y_l^P\right\|_F + \beta^2 r \left\|Y_{l-1}^P\right\|_2^2 \\
=& \beta^2 \left(\left\|Y_{l-1}^P\right\|_F^2 + r \left\|Y_{l-1}^P\right\|_2^2\right) - 2\beta \left((1-\varepsilon)\left\|Y_{l-1}^P\right\|_F \left\|Y_l^P\right\|_F - \sqrt{r}\left\|Y_{l-1}^P\right\|_2 \left\|Y_l^P\right\|_F\right) \\
=& \beta^2 \left(\left\|Y_{l-1}^P\right\|_F^2 + r \left\|Y_{l-1}^P\right\|_2^2\right) - 2\beta \left((1-\varepsilon)\sqrt{\varphi(Y_{l-1}^P)} - \sqrt{r}\right)\left\|Y_{l-1}^P\right\|_2 \left\|Y_l^P\right\|_F,
\end{aligned} \tag{47}$$

where the second inequality is due to Assumption 1.

Taking $r_0 = (1-\varepsilon)^2 \varphi(Y_{l-1}^P)$. If $r \le r_0$, then $\left\|Y_l^P - \widetilde{Y}_{l,\beta}^P\right\|_F^2 \le \left\|Y_l^P - \mathrm{LR}_r(Y_l^P)\right\|_F^2$ holds if

$$\beta = \frac{\left((1-\varepsilon)\sqrt{\varphi(Y_{l-1}^P)} - \sqrt{r}\right)\left\|Y_{l-1}^P\right\|_2 \left\|Y_l^P\right\|_F}{\left\|Y_{l-1}^P\right\|_F^2 + r \left\|Y_{l-1}^P\right\|_2^2}.$$

Thus, we finish the proof of this theorem. $\square$

# E  EXPERIMENTAL DETAILS AND ADDITIONAL EXPERIMENTS

## E.1  EXPERIMENTAL DETAILS FOR THE OBSERVATION OF CROSS-LAYER LOW-RANK STRUCTURE

To empirically validate the cross-layer low-rank structure in different models and training periods., we conducted comparative experiments by fine-tuning the pre-trained LLaMA-3 8B (Grattafiori et al., 2024) and GPT-2-small (Radford et al., 2019) models. And we also pre-training a LLaMA-3 8B

Table 7: Hyperparameter configurations for Qwen-3-based model with MoE architecture.

| Hidden | Intermediate | MoE intermediate | KV Heads | Heads |
|---|---|---|---|---|
| 1024 | 2736 | 704 | 8 | 16 |
| **Layers** | **Experts** | **Activated experts** | **Steps** | **Training tokens (B)** |
| 28 | 24 | 4 | 60K | 7.8 |

Table 8: Hyperparameters for fine-tuning experiments for the low-rank observation.

| Hyperparameters | LLaMA-3 8B | GPT-2 small | Qwen-3 MoE |
|---|---|---|---|
| Optimizer | | AdamW | |
| Learning rate | 1e-5 | 1e-5 | 2.5e-3 |
| Total batch size | 128 | 1024 | 256 |
| Sequence length | 64 | 512 | 256 |
| Warmup iterations | 500 | 500 | 1000 |
| Evaluate every steps | 20 | 10 | 500 |
| Update Steps for fine-tuning | 2000 | 2000 | N.A. |
| Update Steps for pre-training$^\diamond$ | 100 | N.A. | 100 |

$^\diamond$ Here we aim to validate the cross-layer low-rank property in the initial period of pre-training process. Thus we only need to train the model with fewer steps.

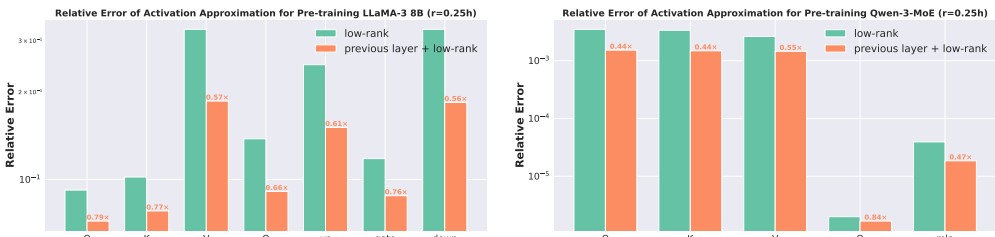

Figure 7: The average relative error of activation recovery by using low-rank approximation and using (3) over all transformer layers. (Left: LLaMA-3 8B, right: Qwen-3-MoE.)

model as well as a model built from Qwen-3-MoE model (Yang et al., 2025). The hyperparameter configurations for Qwen-3-based model is as Table 7. We use the TinyShakespeare dataset for both pre-training and fine-tuning.

We subsequently quantified the activation approximation errors through two distinct methodologies: (a) direct low-rank estimation $\mathrm{LR}_r(Y_l^{\mathrm{P}})$ and (b) matrix recovery via Equation (3). The rank parameter $r = 0.25h, 0.5h$ respectively. For $\widetilde{Y}_{l,\beta_0}^{\mathrm{P}}$ defined in Eq. (3), the scaling factor $\beta_0$ is selected from the set $\{0, 0.2, 0.4, 0.6, 0.8, 1, 2, 3, 5\}$ to minimize the relative error given by (4). And the other hyperparameters can be referred from Table 8.

Figure 8 also presents a detailed relative error of activation approximation for each layer in fine-tuning tasks. For LLaMA-based models, the linear layers at position V, up, down suffer from a higher approximation error, calling for a higher rank to enhance the model performance. For GPT-based models, the linear layers at position V and down-projection suffer from a higher approximation error than the other layers.

Furthermore, Figure 7 illustrates the average relative error of activation recovery by using low-rank approximation and using (3) over all transformer layers in pre-training tasks. Same as the observation in Section 3.1, it can be observation a uniform advantage for estimating activation by the previous layer and a low-rank approximation of the difference between the layers, which illustrates that the cross-layer low-rank structure for activation exists in the initial stage of training.

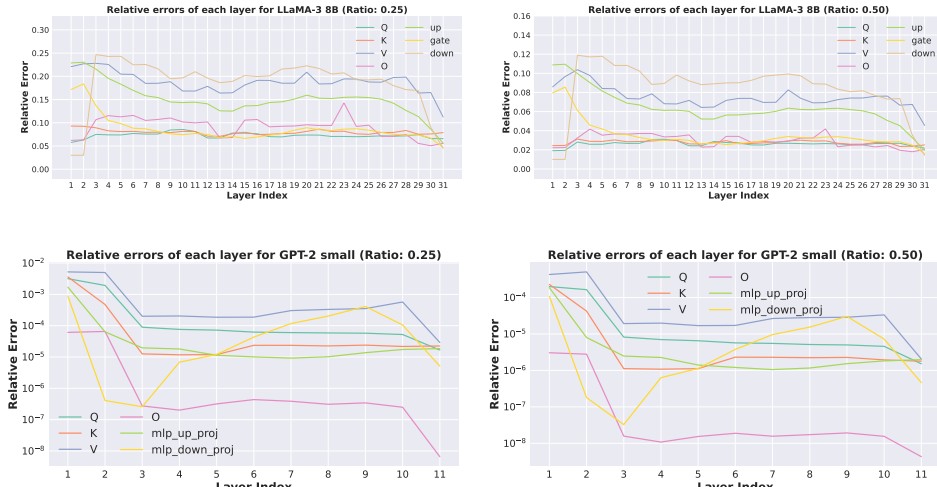

Figure 8: The relative error of activation approximation by using (3) for each transformer layers. 'Ratio' stands for the ratio between the low-rank dimension $r$ and the hidden dimension $h$. (Top: LLaMA-3 8B, bottom: GPT-2 small.)

Table 9: Hyperparameter configurations for LLaMA-2 models of different scales, along with the corresponding number of training steps.

| Parameters | Hidden | Intermediate | Heads | Layers | Steps | Training tokens (B) |
|---|---|---|---|---|---|---|
| 60M | 512 | 1376 | 8 | 8 | 10K | 1.3B |
| 130M | 768 | 2048 | 12 | 12 | 20K | 2.6B |
| 350M | 1024 | 2736 | 16 | 24 | 60K | 7.8B |
| 1B | 2048 | 5461 | 32 | 24 | 100K | 13.1B |
| 7B | 4096 | 11008 | 32 | 32 | 150K | 19.7B |
| 13B | 5120 | 13653 | 40 | 40 | 200K | 26.2B |

Table 10: Layer rank selections for pre-training LLaMA models with *CR-Net* under the scenario of aligned parameters (marked as $\Diamond$ in Table 3) and aligned memory (marked as † in Table 3).

| Parameters | Rank for aligned parameters$^{\Diamond}$ | Rank for aligned memory$^{\dagger}$ |
|---|---|---|
| 60M | Layer 2-4: 96. Layer 5-8: 112. | Layer 2-4: 96. Layer 5-8: 112. |
| 130M | Layer 2-4: 192. Layer 5-12: 224. | Layer 2-4: 192. Layer 5-12: 256. |
| 350M | Layer 2-16: 224. Layer 17-24:256. | Layer 2-24: 384. |
| 1B | Layer 2-24: 448. | Layer 2-24: 768. |
| 7B | Layer 2-32: 896. | N.A. |
| 13B | Layer 2-40: 1260. | N.A. |

## E.2 EXPERIMENTAL DETAILS FOR C4 PRE-TRAINING TASKS

### E.2.1 BASIC EXPERIMENTAL SET UP

During pre-training across all LLaMA model scales, we implement the standardized configuration framework from (Zhao et al., 2024), with key technical specifications comprising a 256-token maximum sequence length and a global batch size of 512 samples, translating to 13.1K tokens per batch. The learning rate scheduling integrates two-phase optimization: initial linear warm-up during the first 10% of training iterations, succeeded by cosine decay gradually reducing the learning rate to 10% of its initial magnitude. Complete architectural configurations and training protocol details are systematically documented in Table 9.

Table 11: Comparison of validation perplexity (↓) of different approaches in LLaMA-2 1B pre-training tasks. The results of compared methods are referred from (Liu et al., 2025; Zhu et al., 2024).

| Training tokens (B) | 3.5 | 10.7 | 13.1 |
|---|---|---|---|
| LoRA | N.A. | N.A. | 19.21 |
| CoLA | N.A. | N.A. | 15.52 |
| *CR-Net* | 19.20 | 15.52 | 15.22 |

Table 12: Comparison of validation perplexity (↓) of different approaches in LLaMA-2 13B pre-training tasks.

| Steps | 10K | 20K | 30K | 40K |
|---|---|---|---|---|
| Full-rank with 8-bit Adam | 24.31 | 20.53 | 18.84 | 17.85 |
| *CR-Net* with re-computation | 25.99 | 21.73 | 19.32 | 18.12 |

As for *CR-Net*, we turn the learning rate over $\{0.0003, 0.0006, 0.001, 0.005, 0.01, 0.012, 0.015, 0.02\}$ and select the learning rate that achieve the lowest validation perplexity over 10% training steps for the whole training process. For low-rank parameter matrices, the learning rate is multipled by 0.25 for scaling. The selection of ranks of *CR-Net* when training with aligned parameter and aligned memory are listed in Table 10.

**Experimental setup for training and inference throughput evaluation.** To measure throughput performance, we pre-trained the LLaMA-2 1B model on an A100 40G GPU with a batch size of 16. For *CR-Net* , we evaluated both single-GPU configurations and multi-GPU scenarios employing data parallelism across 4 GPUs. For inference throughput measurements, we utilized a A100 80G GPU with a microbatch size of 64. Additionally, when assessing throughput for the LLaMA-2 7B model, we employed a A100 80G GPU with a batch size set to 512. In all experiments, we utilized the maximum microbatch size that could be accommodated without triggering out-of-memory errors.

### E.2.2 *CR-Net* ACCELERATES THE PRE-TRAINING TASKS

According to the evaluation perplexity results in Table 3, *CR-Net* achieves a perplexity of 15.22 when training the LLaMA-2 1B model on 13.1B tokens, representing an approximately 2% improvement over both full-rank training and CoLA. Regarding pre-training acceleration, a comparison of validation perplexity during the training of LLaMA-2 1B is provided in Table 11. *CR-Net* reaches the same perplexity as LoRA and CoLA with only 29.02% and 81.52% of their training steps, respectively. When considering the per-step computational FLOPs listed in Table 19, *CR-Net* achieves a $\mathbf{1.326\times}$ acceleration over CoLA and a $\mathbf{18.489\times}$ acceleration over LoRA.

### E.2.3 SCALING *CR-Net* TO A LARGER MODEL SIZE

To validate the performance of *CR-Net* with a larger model size, we pre-train a LLaMA-2 13B model under 8-bit Adam and *CR-Net* separately. We use the re-computation present in Section 3.3 for *CR-Net* and save a series of full activation every 8 layers. We trained 40K steps due to the time limitation. The hyperparameter $r$ in *CR-Net* is set to 1260.

The evaluation perplexity shown as Table 12 illustrates the performance of *CR-Net* in 13B model, which demonstrates that *CR-Net* achieves more than 50% reduction in parameters, while incurring only a 2% degradation in validation performance.

### E.2.4 PRE-TRAINING *CR-Net* WITH ACTIVATION QUANTIZATION

To validate the performance of *CR-Net* with activation quantization, we pre-train a LLaMA-2 350M model under activation quantization. We adopt the same quantization scheme as Q-LoRA Dettmers et al. (2023), which employs NF4 precision and the DoubleQuant strategy with a block size of 16. Table 13 presents the evaluation perplexity of CR-Net with and without activation quantization, compared to full-rank training. The results indicate that *CR-Net* with 4-bit activation quantization achieves additional activation memory savings while incurring less than a 3% increase in perplexity.

Table 13: Comparison of validation perplexity ($\downarrow$) for *CR-Net* with and without activation quantization in LLaMA-2 350M pre-training tasks.

| Training tokens | 6.4B |
|---|---|
| Full-rank | 18.80 |
| *CR-Net* | 18.95 |
| *CR-Net* with activation quantization | 19.32 |

Table 14: Comparison of validation perplexity ($\downarrow$) and accuracy ($\uparrow$) of fine-tuning tasks with different approaches.

| Dataset | Wikitext | | ArXiv | |
|---|---|---|---|---|
| | perplexity | accuracy | perplexity | accuracy |
| GaLore | 21.98 | 40.86% | 25.17 | 40.59% |
| *CR-Net* | 20.66 | 41.14% | 24.48 | 41.00% |

### E.2.5  DOWNSTREAM PERFORMANCE

To examine the downstream performance of our proposed framework, we fine-tune LLaMA-2 1B models pre-trained by GaLore and *CR-Net* for 15.6B tokens. The training set up is the same as that in the manuscript. The fine-tune datasets consist of Wikitext-2 dataset (Merity et al., 2016) and 30K arXiv abstracts (Wang et al., 2022). We tune each data for 8 epochs. Table 14 presents the evaluation perplexity and accuracy of these tasks, which illustrates that *CR-Net* has better performance than GaLore in these downstream tasks.

### E.2.6  EXPERIMENTAL SETTING AND ADDITIONAL RESULTS FOR THE ABLATIONS

In this section, we present the experimental setting and additional results for the ablations we presented in Section 5.2.

**How does rank selection impact pre-training performance?** In this ablation, we train LLaMA-2 350M with *CR-Net* for 60K iterations. The max learning rate is set to 0.01 and all the other configs is the same as that in Appendix E.2.1. For the selection of layer ranks, we present the following three strategies to ensure the their parameters are all the same:

- **S1:** $r = 256$ in Layer 2-8. $r = 192$ in Layer 9-24.
- **S2:** $r = 256$ in Layer 10-16. $r = 192$ in Layer 2-9 and Layer 17-24.
- **S3:** $r = 256$ in Layer 18-24. $r = 192$ in Layer 2-17.

**Whether does the learnable scale factor $\beta_l^{\mathrm{P}}$ benefit the model convergence?** In this ablation, we use *CR-Net* to train LLaMA-2 350M and LLaMA-2 1B model with a learnable $\beta_l^{\mathrm{P}}$ and a series of fixed $\beta_l^{\mathrm{P}}$ for $l = 2, 3, \cdots, L$ and P $\in \{$Q, K, V, O, gate, up, down$\}$, respectively. The fixed $\beta_l^{\mathrm{P}}$ varies from $0.1, 0.2, 0.5, 1.0, 2.0$ for LLaMA-2 350M and $\beta_l^{\mathrm{P}} = 1.0$ for LLaMA-2 1B. We trained 40000 iterations for LLaMA-2 350M and 80000 iterations for LLaMA-2 1B, respectively. Figure 9 illustrates the evaluation perplexity of each cases, shown the benefit of letting $\beta_l^{\mathrm{P}}$ learnable as the higher validation performance.

**Whether other cross-layer residual strategies do well in pre-training with low-rank parameters?** To compare *CR-Net* with different efficient cross-layer residual strategies for LLMs pre-training including ResFormer (Zhou et al., 2024) and DenseFormer (Pagliardini et al., 2024), we pre-trained a LLaMA-2 1B model with *CR-Net*, Learnable-ResFormer, and DenseFormer using low-rank parameters. The rank $r$ for transformer layers (except the first layer) is set to 448, while all other configs are the same as those in Appendix E.2.1. We stopped training early at the 50,000-th step.

Figure 6 illustrates the evaluation perplexity of *CR-Net* and other cross-layer residual strategies for the LLaMA-2 1B training task. It can be observed that *CR-Net* outperforms the ResFormer architecture and DenseFormer in pre-training with low-rank parameters.

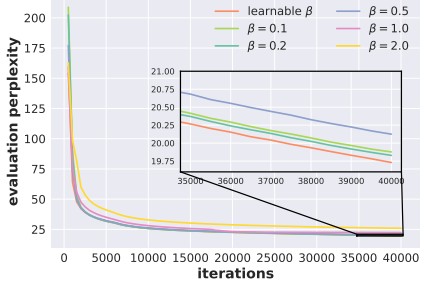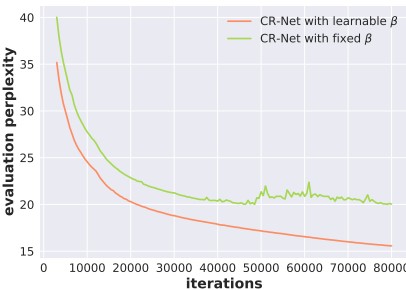

Figure 9: The Comparison of evaluation perplexity for *CR-Net* in pre-training tasks with fixed $\beta_l^{\mathrm{P}}$ and learnable $\beta_l^{\mathrm{P}}$. (Left: LLaMA-2 350M, right: LLaMA-2 1B.)

Table 15: Comparison of validation perplexity ($\downarrow$) of different approaches in LLaMA-2 pre-training tasks with longer sequence length and more training tokens.

| Model size | 130M | 350M | 1B |
|---|---|---|---|
| Training tokens (B) | 2.9 | 8.0 | 29.5 |
| GaLore | 25.14 | 18.87 | 15.03 |
| *CR-Net* | 23.73 | 18.86 | 14.79 |

### E.2.7 PRE-TRAINING WITH LONGER SEQUENCE AND MORE TRAINING TOKENS

To align with the basic experimental setup of existing efficient frameworks for LLMs pre-training like LoRA and GaLore, we follow the setup in Zhao et al. (2024) and use a short sequence length, to validate the pre-training performance in long training tokens as well as more training tokens. We pre-trained LLaMA-2 models using the C4-en dataset with maximum sequence length set to 2048. The total training tokens are more than $22\times$ the parameter size to follow the Chinchilla scaling law (Hoffmann et al., 2022). As shown in Table 15, *CR-Net* outperforms GaLore in this scenario.

### E.3 PRE-TRAINING GQA AND MoE MODELS WITH *CR-Net*

In this section, we present additional experiments on pre-training the LLaMA-3 model (Grattafiori et al., 2024) with a grouped query attention (GQA) (Ainslie et al., 2023) architecture, as well as pre-training a Qwen-3-based (Yang et al., 2025) model with a mixture of experts (MoE) architecture, to validate the proposed framework across different model structures.

**Pre-training LLaMA-3 with GQA architecture.** We pre-trained a 1B-parameter LLaMA-3 model on 15.6B tokens from the C4-en dataset using both GaLore and *CR-Net* methods. The model hyperparameters are shown in Table 16. The low-rank coefficient rr was set to 512 for GaLore and 448 for *CR-Net*. The remaining configuration follows the same setup as the LLaMA-2 pre-training task described in Appendix E.2. The evaluation perplexity results are presented in Table 17, indicating that *CR-Net* outperforms GaLore in pre-training tasks on the LLaMA-3 model.

**Pre-training models with MoE architecture.** We pre-trained a Qwen-3-based model under the MoE architecture, containing 1.8B total parameters with 650M activated parameters, on 7.8B tokens from the C4-en dataset. Training was conducted under both full-rank and *CR-Net* settings. The model hyperparameters are provided in Table 7. The low-rank coefficient rr in *CR-Net* was set to 224. For *CR-Net*, we replaced the parameter matrices within the linear operators of the MoE layers. Each MoE layer was treated as a unified operator, and cross-layer residual connections were applied after obtaining the output of the MoE operators. The remaining experimental setup matches that of the LLaMA-2 pre-training task in Appendix E.2. The evaluation perplexity, shown in Table 18, demonstrates the competitive performance of *CR-Net* compared to full-rank training.

Table 16: Hyperparameter configurations for LLaMA-3 model.

| Parameters | Hidden | Intermediate | KV Heads | Heads | Layers |
|---|---|---|---|---|---|
| 8B | 2048 | 7168 | 8 | 32 | 32 |

Table 17: Comparison of validation perplexity ($\downarrow$) of different approaches in LLaMA-3 pre-training tasks.

| | 40K | 80K | 100K | 110K |
|---|---|---|---|---|
| GaLore | 19.29 | 16.89 | 16.47 | 16.40 |
| *CR-Net* | 18.29 | 16.05 | 15.70 | 15.65 |
| Training tokens (B) | 5.2 | 10.5 | 13.1 | 14.4 |

## F    ADDITIONAL DISCUSSIONS

### F.1    PER-STEP COMPUTATION ACCELERATION ANALYSIS.

To illustrate the computation efficiency of *CR-Net* in practical training tasks, we provide the computation complexity of different efficient pre-training approaches for one gradient step (based on LLaMA-2 architecture) in Section 5.1 as Table 19. It can be observed that *CR-Net* achieves the same training performance as full-rank training with 36.8% computation overhead for both LLaMA-2 1B and 7B models.

### F.2    ADDITIONAL DISCUSSION FOR THE LEARNABLE SCALING FACTOR $\beta_l^{\mathrm{P}}$.

The layer-specific learnable scaling factor $\beta_l^{\mathrm{P}}$ serves as a critical dynamic balancing mechanism between historical information propagation and new low-rank feature generation, rather than a conventional hyperparameter. When exceeds appropriate ranges (e.g., $\beta_l^{\mathrm{P}} > 2.0$), low-rank features dominate current activations, whereas values approaching zero excessively prioritize historical information - both scenarios are demonstrated in Figure 5 to adversely impact model performance.

Notably, $\beta_l^{\mathrm{P}}$ exhibits multidimensional adaptability requirements: optimal values vary across layers, spatial positions, and temporal phases of training. This intrinsic variation motivates our proposal for learnable $\beta_l^{\mathrm{P}}$ parameterization. As evidenced by Figure 5 and 6, dynamic $\beta_l^{\mathrm{P}}$ adaptation yields consistent performance improvements across model scales compared to static configurations.

To examine the empirical value distribution for beta terms, we obtain the distribution of all $\beta_l^{\mathrm{P}}$ terms in a pre-trained LLaMA-2 350M model with 20,000 steps. As shown in Table 20, the $\beta_l^{\mathrm{P}}$ terms mostly range from 0.20 to 1.00, where the model may not suffer from the negative impact of extreme betas.

## G    MULTI-GPU TRAINING: AN ANALYSIS FOR *CR-Net* WITH PIPELINE PARALLELISM

While the single-GPU throughput results (Figure 3) validate the computational efficiency of *CR-Net* , multi-GPU training—especially with parallelism strategies like Pipeline Parallelism (PP) (Narayanan et al., 2019; 2021; Huang et al., 2019; Ryabinin et al., 2023)—introduces critical considerations around communication overhead. Cross-layer residual connections, a core component of *CR-Net*, could theoretically increase data transmission between GPUs, as forward/backward passes require sharing activation residuals across worker nodes. To address this concern, we conduct a **quantitative analysis of computation-communication tradeoffs** for *CR-Net* in PP-enabled multi-GPU setups as well as an empirical verification. We also present an analysis of HBM memory with pipeline parallelism.

**Basic setup.** We conducted a pre-training task of *CR-Net* with re-computation and full-rank training with vanilla in pipeline parallel deployment on A100 80G GPUs for LLaMA-2 models with 13B

Table 18: Comparison of validation perplexity ($\downarrow$) of different approaches in Qwen-3-based pre-training tasks.

|  | 10K | 20K | 30K | 40K |
|---|---|---|---|---|
| Full-rank | 24.31 | 20.53 | 18.84 | 17.85 |
| *CR-Net* | 25.99 | 21.73 | 19.32 | 18.12 |
| Training tokens (B) | 1.3 | 2.6 | 3.9 | 5.2 |

Table 19: Computation complexity of different efficient pre-training approaches for one gradient step based on LLaMA architecture for the task in Section 5.1. Lower-order terms are omitted for brevity. The selection of the rank $r$ for *CR-Net* are based on Table 10 while that of the other algorithms are based on (Zhao et al., 2024; Liu et al., 2025).

| Approach | LLaMA-2 350M | LLaMA-2 1B | LLaMA-2 7B |
|---|---|---|---|
| Full-rank | $4.838 \times 10^{11}$ (1.000$\times$) | $2.525 \times 10^{12}$ (1.000$\times$) | $1.005 \times 10^{13}$ (1.000$\times$) |
| (Re)LoRA | $8.128 \times 10^{11}$ (1.674$\times$) | $4.226 \times 10^{12}$ (1.673$\times$) | $1.678 \times 10^{13}$ (1.670$\times$) |
| SLTrain | $9.482 \times 10^{11}$ (1.960$\times$) | $7.473 \times 10^{12}$ (2.959$\times$) | $4.984 \times 10^{13}$ (4.959$\times$) |
| GaLore | $7.934 \times 10^{11}$ (1.640$\times$) | $5.824 \times 10^{12}$ (2.306$\times$) | $3.658 \times 10^{12}$ (3.639$\times$) |
| CoLA | $1.934 \times 10^{11}$ (0.400$\times$) | $1.005 \times 10^{12}$ (0.398$\times$) | $0.398 \times 10^{13}$ (0.396$\times$) |
| *CR-Net* | $1.985 \times 10^{11}$ (0.410$\times$) | $0.930 \times 10^{12}$ (0.368$\times$) | $0.369 \times 10^{13}$ (0.367$\times$) |

Table 20: Distributions of beta values for a pre-trained LLaMA-2 350M model.

| Beta | $[0.00, 0.20)$ | $[0.20, 0.60)$ | $[0.60, 1.00)$ | $[1.00, 1.25)$ |
|---|---|---|---|---|
| Frequency | 0.05625 | 0.2625 | 0.55 | 0.13125 |

and 70B parameters. We assume the intermediate dimension of FFN layers is $8/3$ times the hidden dimension. The other model hyperparameters are listed in Table 21.

For *CR-Net*, we set the low-rank coefficient $r = 0.25h$ and save the entire series of activations every 8 layers for re-computation. We ignore higher-order computation overhead in the complexity analysis. Following the analysis in our manuscript, the computation complexity of *CR-Net* and full-rank training for one gradient step can be obtained.

**Communication overhead.** Then, when using BF16 precision, the communication overhead can be computed by:

$$\text{Volume (GBytes)} = \frac{\text{Microbatch} \times \text{Communication dimension} \times 2 \times 3}{10^9}, \tag{48}$$

where the multiplication by 2 is due to the 2-byte requirement for storing each parameter in the BF16 precision, and the multiplication by 3 accounts for the communication overhead incurred during the forward pass, backward pass, and re-computation [1].

Using the peak computation performance of A100 with BF16 (312 TFLOPS) and PCIe 4.0 bandwidth (64 GB/s), the computation and communication times per gradient step are shown in Table 22. Notably, *CR-Net*'s computation time savings are overwhelmingly significant—exceeding about 30 times of the additional communication time for 13B and 70B models respectively—**far outweighing the minimal communication overhead**. Moreover, when using a GPU-connection strategy with higher bandwidth (e.g., NVLink), the communication time can be further reduced, allowing computation time to dominate the training process even more significantly. This results in greater total time savings due to *CR-Net*'s computational efficiency. Since inference mirrors the forward pass of pre-training, these findings directly validate that *CR-Net*'s computation efficiency gains surpass trivial communication costs, confirming its superiority in multi-GPU deployment.

---

[1] For full-rank training, if we split the model as the end of a transformer layer, then the re-communication may not lead to communication overhead here.

Table 21: Hyperparameter configurations for LLaMA-2 models of different scales, along with the corresponding number of devices for a propagation pipeline. 'Microbatch' denotes the size of one microbatch. 'PP size' number of devices for a GPU pipeline for the forward- and backward-propagation of one microbatch.

| Parameters | Hidden | Heads | Layers | Sequence length | Microbatch | PP size |
|---|---|---|---|---|---|---|
| 13B | 5120 | 40 | 40 | 4096 | 16 | 2 |
| 70B | 8192 | 64 | 80 | 4096 | 16 | 8 |

Table 22: Computation and communication times for one gradient step. *CR-Net* achieves time saving of 13.42s and 127.63s in 13B and 70B training tasks per step, respectively.

| Parameters | Methods | Computation FLOPs ($\times 10^{15}$) | Computation time (s) | Communication overhead (GB) | Communication time (s) |
|---|---|---|---|---|---|
| 13B | Full-rank | 7.48 | 23.97 | 1.88 | 0.029 |
| | *CR-Net* | 3.19 | 10.23 (**13.74-**) | 22.5 | 0.352 (**0.323+**) |
| 70B | Full-rank | 36.67 | 177.53 | 21.00 | 0.328 |
| | *CR-Net* | 14.51 | 46.51 (**131.02-**) | 238.0 | 3.719 (**3.391+**) |

**Empirical verification.** To empirically validate communication efficiency, we pre-trained a LLaMA-2 70B model with CR-Net under pipeline parallelism using 8 NVIDIA A800 GPUs interconnected via NVLink, on the C4-en dataset (sequence length=256). Pipeline parallel training was implemented with the `torch.distributed.pipelining` library, with each GPU handling 10 transformer layers.

The measured communication time for one forward and backward step was **1.380 seconds**, yielding a bandwidth of 192.5 GB/s. This bandwidth is approximately three times higher than the PCIe bandwidth assumed in our theoretical analysis (Table 22). These findings demonstrate that under NVLink connectivity, *CR-Net* achieves shorter communication times than theoretically projected, further confirming that for large scale models, the communication volume does not substantially impair overall efficiency, thus supporting our theoretical conclusions.

**HBM memory.** We also clarify that *CR-Net*'s low-rank parameterization inherently reduces overall HBM memory requirements. For the pre-training tasks for LLaMA-2 70B with BF16 precision, under the configuration outlined in Table 21, *CR-Net* introduces 11.3GB additional activation memory per device versus 1GB in full-rank training. Meanwhile, *CR-Net* achieves about 50% reduction in memory for parameters, gradients, and optimizer states, yielding a net 29.6GB HBM saving per device. This memory efficiency enables LLM training with fewer devices, fundamentally alleviating both communication overhead and collective HBM demands.

## LLMS USAGE.

In this paper, generative LLMs were used solely for writing polishing, such as grammar and wording improvements. All LLM-edited content was manually verified to ensure compliance with ICLR policies, and authors bear full responsibility for the submission.

