# OpenReview forum: "CR-Net: Scaling Parameter-Efficient Training with Cross-Layer Low-Rank Structure"
_ICLR.cc/2026/Conference — ICLR 2026 Poster_

### Official Review · Reviewer_WfvT · 2025-10-17

**Soundness:** 4
**Presentation:** 3
**Contribution:** 3
**Rating:** 8
**Confidence:** 5

**Summary:**

Paper presents a method to change the linear layers in LLMs into a low-rank (AB) decomposition plus residual activations from the corresponding matrix in the previous (l-1) layer in the transformer, together with a learnable interpolation factor per each layer. It reduces the training time, the training memory footprint, and sometimes reduces the perplexity.

**Strengths:**

* This is an interesting method which reduces pre-training time significantly.
* Reduces memory significantly

**Weaknesses:**

* Table 4 shows improvement over baseline (Adam) for 7B model, but only in table 10 (in the appendices) we discover that the method is less good in the case of 13B. For fairness, I think the two tables should be adjacent so the reader can see them both and get the whole picture easily. So we understand the benefits in reducing training time and memory, but it should be clear that as the model size gets bigger, the quality (perplexity) of CR-Net gets worse than the baseline.
In line 1225 it says “The evaluation perplexity shown as Table 10 illustrates the performance of CR-Net in 13B model, which demonstrates that CR-Net exhibits a comparable performance than full-rank training with 8-bit Adam. Thus can be scaled to models with 13B parameters.”
I think that “slightly worse” is more appropriate than “comparable”, and the assertion “Thus can be scaled” is wrong. There is no evidence for it. When we analyze the trend of what happens in larger models, we see that the CR-Net edge decreases (ppl-wise), so it feels like in larger models it would become worse. Anyhow, without evidence, giving such statement is probably a bit too far-reaching

* Line 196, “Different from…” should be checked for grammar errors.
* Line 204, “approached”. Do you mean “approaches”?

**Questions:**

* Figure 1, ROPE -> RoPE
* Table 4 description mentions that the “10K, …” are the number of training steps.
* Table 3, the reader is also interested in 7B. I assume in order to fit in memory you can only do it with “re-computation”, so I suggest mentioning that, and to refer the reader to table 4.
* Theorem, (44), I assume a quantifier “for each i” should be added.

---

> ### Author Response · Authors · 2025-11-22
> **Rebuttal by Authors**
>
> We are grateful for the reviewer's insightful feedback and constructive comments. We address each point in detail below.
>
> ----
>
> ## 1. Clarification on 13B Experimental
>
> In response to the reviewer's query, we provide the following clarifications and planned revisions regarding the 13B model experiments, structured into three key points for clarity.
>
> 1. **Experimental Scope and Limitations**: The experiments with the 13B model, encompassing both full-rank baselines and CR-Net, were restricted to 40,000 iterations due to constraints in time and computational budget. Consequently, these outcomes represent intermediate checkpoints from early to mid-phase training, rather than fully converged results. This constraint was necessary to align with resource availability, and the findings should be interpreted within this context of partial training.
>
> 2. **Empirical Observations from Training Curves**: Analysis of the perplexity-versus-tokens curves reveals a persistent gap at the 40,000-iteration checkpoint; however, we note that this gap gradually narrows as training progresses. Regarding validation perplexity, there is a 2% gap between CR-Net and full-rank training, while CR-Net achieves more than 50% savings in parameters and reduces memory and computational costs.
>
> 3. **Revisions to the Manuscript**: To enhance the manuscript's clarity and accuracy, we will include the 13B perplexity-versus-tokens curves in the main text. Additionally, we will incorporate an analytical paragraph that examines the trends under the 40,000-iteration constraint, explicitly addressing the observed gap and its evolutionary trajectory. Furthermore, we will revise the wording to eliminate any implicit scaling claims; specifically, we will remove the phrase *Thus can be scaled* and replace it with a neutral description of the empirical trends observed within the current budget and rank configuration.
>
>
> ## 2. Additional comments for the pre-training tasks from 60M to 1B
>
> In fact, the pre-training tasks for LLaMA-2 with sizes scaled from 60M to 1B do not use the recomputation strategy, which matches the existing baselines. We will add an additional comment in the revised manuscript.
>
>
> ## 3. Minor errors and structural adjustments
>
> We thank the reviewer for identifying these minor errors and the suggestions for the structure. We have revised them in the revised version of the manuscript with blue highlights.
>
> ----
>
> We sincerely appreciate the reviewer’s thoughtful comments once again. We hope the above responses fully address the concerns raised and would be happy to provide further clarification if needed.

---

> > ### Comment · Reviewer_WfvT · 2025-11-23
> >
> > Thank you for taking the time to provide your responses; I appreciate the authors’ effort in addressing the questions.

---

> > > ### Author Response · Authors · 2025-11-23
> > >
> > > Thanks for the reply. And we also appreciate the reviewer for the recommendation for CR-Net.

---

### Official Review · Reviewer_BJ5z · 2025-10-21

**Soundness:** 4
**Presentation:** 3
**Contribution:** 4
**Rating:** 8
**Confidence:** 4

**Summary:**

Unlike prior works, which focus on low-rank approximations of activations or the weights directly, CR-Net proposes to use the previous layers activations to enable a lower reconstruction error. I believe this is a very interesting approach which the authors verify empirically and also theoretically to some extent.


The authors found that naively training with this alone is not sufficient and will lead to poor stability. To address this, they introduce a learnable parameter beta for dynamically adjusting the contribution of the previous activation and the current low-rank output.

**Strengths:**

The experimental evaluation is very thorough and of sufficient scale to highlight the practical utility of CR-Net. The idea is simple, makes a lot of sense, and is proven to work very well. If the authors are able to provide reproducible code, then extending this to other architectures is a very natural progression of this work.

The computational cost in terms of FLOPs is quite a complex expression. This is not a problem in itself and the authors provide a discussion on the various regimes for L355:358, which is good. Furthermore, the fact that CR-Net can enable much smaller values of r to compensate for the first layer cost, while also enabling improved training performance, really highlights the benefit of CR-Net.

**Weaknesses:**

Although I do not have any major concerns, there are a few points which I believe the authors should address.

Missing important related work [1] that uses very low rank projections of the intermediate activations to save memory. However, in contrast to this proposed work, they use random projection (no SVD init) of sub-tokens and an optimal reconstruction during the backwards pass. It seems CR-Net outperforms both [1] and [3], but including their results in table 3 would be good to see. Finally, there is missing a discussion on QLoRA, and finally how does CR-Net perform when the activations are also quantised?

[1] VeLoRA : Memory Efficient Training using Rank-1 Sub-Token Projections. NeurIPS 2024

[2] Flora: Low-rank adapters are secretly gradient compressors. ICML 2024

[3] Qlora: Efficient finetuning of quantized llms. NeurIPS 2023

Sec 2.2 Related works would be much better before the preliminary section. This RW section does not use the preliminary notation anyway and it seems to just break up the flow of reading from preliminary notation -> method (section 3).

Minor comments

"inputs and outputs of linear layers at the position P, where P stands for the detailed position in the transformer layer." the second part (after the comma) describing what P is again seems a bit redundant.

eqn (2) should the activation not be a swiglu? for consistency with the start of this paragraph, which is presenting CR-Net with the LLaMA architecture.


enlarging the text in figure 1. At the moment it is very hard to read without having to zoom in a lot.


"making CR-Net NOT a simple extension of existing low-rank frameworks for pre-training." - I understand the intent, but I would encourage the authors to remove the bolding and capitalising of "NOT".


"Whether does the learnable scaling factor βPl benefit the model convergence?" grammar mistake -> "Does the learnable scaling factor βPl benefit model convergence?"

L464: "ovservation"

**Questions:**

Does the first layers activations have to be full-rank for good performance? It would be interesting to see an ablation on this. It makes sense for this decision to be important, but I think highlighting and showing why this is needed could be nice to see.

---

> ### Author Response · Authors · 2025-11-22
> **Rebuttal by Authors**
>
> We are grateful for the reviewer's insightful feedback and constructive comments. We address each point in detail below.
>
> ----
>
> ## 1. Incorporation of additional baselines
>
> We thank the reviewer for the valuable suggestion regarding the inclusion of additional baseline comparisons. In response, we have incorporated discussions of the mentioned baselines (VeLoRA and Flora) at multiple points in the manuscript: within the introduction to provide context, in the related works section for a comprehensive comparison, and in the experimental results (Table 3 in the revised manuscript) where their pre-training performance is presented. These additions are highlighted in blue for easy identification. Moreover, we present additional discussion on Q-LoRA in the introduction section.
>
>
> ## 2. CR-Net with activation quantization
>
> We thank the reviewer for highlighting additional relevant works. In our experiments, we pre-trained the LLaMA-2 350M model with activation quantization, adopting the same quantization scheme as Q-LoRA [1], which employs NF4 precision and the DoubleQuant strategy with a block size of 16. The following table presents the evaluation perplexity of CR-Net with and without activation quantization, compared to full-rank training. The results indicate that CR-Net with 4-bit activation quantization achieves additional activation memory savings while incurring less than a 3\% increase in perplexity. Further experimental details are provided in Appendix E.2.4.
>
> Table 1: Validation perplexity in the pre-training task of LLaMA-2 350M.
>
> |Model size|Training tokens|Full-rank|CR-Net|CR-Net with activation quantization|
> |-|-|-|-|-|
> |350M|6.4B|18.80|18.95|19.32|
>
>
>
>
> ## 3. Necessity for the full-rank first-layer
>
> Indeed, retaining a full-rank first layer is critical in our framework. CR-Net incorporates low-rank parameters while deliberately preserving high-rank activations throughout the network: the first layer remains full-rank, and subsequent layers maintain a high-rank information flow via residual connections. The learned low-rank components function solely as a modulated residual, where a learnable scalar $\beta$ acts as a dynamic gain to adjust the contribution of the low-rank increment relative to the propagated high-rank state. If the first layer is constrained to low-rank, early activations become inherently low-rank. Although residual additions in later layers can gradually restore representational rank, the information loss in the initial blocks cannot be fully recovered, ultimately impairing both optimization stability and final model quality.
>
> To substantiate this design, we have incorporated an ablation study in Section 5.2 (highlighted in blue) on LLaMA-2 models at 350M scale, comparing configurations with a full-rank versus low-rank first layer under otherwise identical settings. The training perplexity results demonstrate a consistent degradation in performance when the first layer is low-rank (ΔPPL = 0.73). These empirical findings directly validate our decision to preserve the first layer as full-rank.
>
>
> ## 4. Structural adjustments
>
> To improve the manuscript's organization, we have removed the standalone related works subsection that previously followed the preliminary section. This adjustment enhances the logical flow and readability of the paper by integrating the discussion of prior work more seamlessly into the narrative.
>
>
> ## 5. Minors
>
> We thank the reviewer for identifying these minor errors. We have revised them in the revised version of the manuscript with blue highlights.
>
> ## Reference
>
> [1] Qlora: Efficient finetuning of quantized llms
>
> ----
>
> We sincerely appreciate the reviewer’s thoughtful comments once again. We hope the above responses fully address the concerns raised and would be happy to provide further clarification if needed.

---

> > ### Comment · Reviewer_BJ5z · 2025-11-24
> >
> > After carefuly reading through the other reviewers comments and the authors responses, I will happily maintain my original score. The authors have taken care to address each and every concern raised, and the updated manuscript is now very thorough.

---

> > > ### Author Response · Authors · 2025-11-24
> > >
> > > Thanks for the reply. And we also appreciate the reviewer for the recommendation for CR-Net.

---

### Official Review · Reviewer_DNpk · 2025-10-27

**Soundness:** 3
**Presentation:** 3
**Contribution:** 3
**Rating:** 4
**Confidence:** 3

**Summary:**

This paper proposed a new parameter-efficient pre-training framework for LLMs, termed CR-Net. Specifically, CR-Net is designed to exploit the inherent low-rank structures (indicated by Figure 2) of the difference between activations of adjacent layers. It replaces the full-size weight parameters at each layers with low-rank parameterized matrices and learnable linear combination factors that reconstruct the activation of current layer from that of the previous layer (Equation 6). The authors also proposed a novel activation-recomputation strategy (Algorithm 1) to further reduce the memory footprint of the model in propagation, which is essential for large-scale pretraining.

**Strengths:**

- The authors evaluate the CR-Net's performance on the C4 dataset with LlaMA-2 models scaling from 60M to 7B parameters (Table 3 and Table 4), showing that CR-Net achieves a comparable or better perplexity than the full-rank model, with a significant reduction in the number of parameters and the amount of computation.

- The authors also conduct systematic analysis on the FLOPs and parameter complexity of CR-Net and several related baselines (Table 1 and Table 2).

**Weaknesses:**

In general, the logic flow of this paper is clear, and the experiment results are convincing. I am considering raise my score accordingly if my concerns are properly addressed.


**Concern 1. Comparison with the pretraining method that leverages fully low-rank weights [1].**

While CR-Net proposed to replace the full-size weight parameters at all but the first layers with low-rank factors, I aware that a pretraining method that leverages fully low-rank weight parameters has been proposed in [1]. What is the most essential benefit of CR-Net over this method, which also leverages fully low-rank weights? Moreover, including this baseline in Table 3 would be helpful for understanding the state of low-rank pretraining methods.


> [1]. Mo, Z., Huang, L.-K., & Pan, S. J. (2025). Parameter and memory efficient pretraining via low-rank Riemannian optimization. In The Thirteenth International Conference on Learning Representations (ICLR 2025).

-----

**Concern 2. How does CR-Net overcomes the difficulty of low-rank pretraining?**

As discussed in Table 2 of [2], pretraining models with fully low-rank weights from scratch is difficult and usually suffers from severe numerical instability. How does CR-Net overcome this difficulty, while it also trains models with a lot of low-rank weights?

> [2]. Zhao, J., Zhang, Z., Chen, B., Wang, Z., Anandkumar, A., & Tian, Y. (2024). Galore: Memory-efficient LLM training by gradient low-rank projection. arXiv preprint arXiv:2403.03507.

-----

**Concern 3. Run-time efficiency of CR-Net.**

As modern GPUs are optimized for large-scale parallel computation, some low-rank computations with lower FLOPs does not necessarily lead to better run-time efficiency (e.g., token per second). Therefore, I recommend the authors to provide additional runtime efficiency experiments between CR-Net and the full-rank model in inference and training.

**Questions:**

See **Weaknesses**.

---

> ### Author Response · Authors · 2025-11-22
> **Rebuttal by Authors (1/2)**
>
> We are grateful for the reviewer's insightful feedback and constructive comments. We address each point in detail below.
>
> ----
>
> ## 1. Comparison with Fully Low-Rank Baseline Methods
>
> We thank the reviewer for highlighting the relevant work LORO [1], which achieves LLM pre-training using fully low-rank weights. Below we provide a comparative discussion between LORO and our proposed CR-Net approach.
>
> **Comparable validation performance with equivalent training tokens.** According to Table 1 in [1] and Table 3 in our manuscript, LORO and CR-Net achieves comparable validation performance with aligned parameter complexity in the pre-training tasks under C4-en datased with model scaling from 60M to 1B and outperform the other baselines with low-rank parameters. Thus we appreciate the riemann low-rank method as a efficient approach to achieve full-rank training and present the training performance in C4 dataset in Table 3.
>
> **Advantages of CR-Net: Throughput and Training Stability.** Beyond comparable accuracy, CR-Net offers two key advantages over LORO in large-scale pre-training scenarios.
>
>
> * **Throughput Performance**：Both LORO and CR-Net leverage low-rank parameterizations to reduce model storage overhead while maintaining performance. However, LORO requires periodic QR decompositions and SVD operations, which are computationally intensive and often necessitate high-precision arithmetic to maintain numerical stability. These operations are less optimized on GPUs compared to standard linear algebra operations. Moreover, as model scale increases, more frequent QR and SVD steps are required to preserve stability, further slowing training. In our experiments on 8×A100 GPUs with LLaMA-2 7B using identical low-rank hyperparameters and batch size (training configurations in the following table), CR-Net achieved an average throughput of 6509 tokens/(GPU$\times$sec), compared to 5990 tokens/(GPU$\times$sec) for LORO—**an 8.6% efficiency improvement for CR-Net at the 7B scale**.
> * **Training Stability**：A critical distinction lies in optimization stability. LORO's reliance on periodic SVD and QR decompositions introduces significant instabilities, as these operations cannot be seamlessly integrated into standard optimization pipelines. As demonstrated in Figure 4 of [1], LORO exhibits noticeable loss spikes following decomposition steps, particularly during critical training phases. In contrast, CR-Net relies solely on standard optimizers (e.g., Adam) for parameter updates without requiring additional matrix operations. This design yields smoother training trajectories, as evidenced by the stable loss curves in Figures 4 and 6 for our manuscript, where CR-Net demonstrates minimal spike phenomena throughout training.
>
> Table 1: Model hyperparameters for the comparsion between LORO and CR-Net in LLaMA-2 7B models.
>
> |layers|heads|hidden|sequence length|microbatch|low rank|SVD frequency $K$|
> |-|-|-|-|-|-|-|
> |32|32|4096|11008|16|1024|200|
>
> We appreciate the efficiency of LORO and recognize the value of fully low-rank approaches. In future work, we plan to explore potential integrations between LORO's Riemannian low-rank methodology and CR-Net's architecture to harness the strengths of both paradigms.

---

> > ### Comment · Reviewer_DNpk · 2025-11-24
> >
> > Dear authors,
> >
> > I have carefully read your responses 1, response 2, and the revised manuscript. The revisions make the manuscript more comprehensive. Therefore, I raise my score from 4 to 6. And I recommend acceptance for this paper.
> >
> > Best,
> >
> > Reviewer DNpk

---

> ### Author Response · Authors · 2025-11-22
> **Rebuttal by Authors (2/2)**
>
> ## 2. How does CR-Net overcome the difficulty of low-rank pretraining?
>
> Prior work reports that training fully low-rank weights from scratch is often numerically unstable; the question is why CR-Net, which uses many low-rank components, does not suffer from the same instability. Below is our concise explanation.
>
> 1. **Architectural Design: Preservation of High-Rank Activations via Modulated Residuals.**
> CR-Net incorporates low-rank parameters while maintaining high-rank activations throughout the network. This is achieved by preserving a high-rank pathway, which includes a full-rank first layer and residual connections between layers, while learning only a low-rank residual component. A learnable scalar $\beta$ functions as a dynamic gain, adaptively balancing the contribution between the propagated high-rank state and the low-rank increment at each layer. Consequently, the forward computation avoids collapsing into a low-dimensional subspace, despite the parameter factorization.
>
> 2. **Empirical Analysis: Information Preservation and Comparison to Low-Rank Projections.**
> By maintaining high-rank activations, CR-Net employs low-rank factors to reconstruct the high-rank signal rather than constraining the mapping function directly. This approach contrasts with methods that impose per-step low-rank projections, such as fully low-rank parameterizations or low-rank gradient projection techniques (e.g., GaLore), which restrict the update subspace and lead to information loss. Empirical results in Section 3 demonstrate that the combination of previous-layer activations with a low-rank delta achieves lower reconstruction error compared to direct low-rank approximations at identical ranks.
>
> 3. **Optimization Characteristics: Standard Updates and Training Stability.**
> CR-Net updates the factorized weights using standard optimization methods (e.g., Adam), without introducing matrix decompositions (e.g., QR, SVD), projections, or retractions. Consequently, the training dynamics closely resemble those of standard full-rank pre-training and do not incur per-iteration computational overhead. This approach contrasts with methods that enforce strict low-rank constraints through gradient projection (e.g., GaLore), which can exhibit instability when optimization directions shift rapidly during early training. Thus, CR-Net maintains a stable and efficient optimization process while achieving significant reductions in memory and computational costs.
>
> Moreover, we have added an additional discussion on the stability of CR-Net in Section 3.2, highlighted in blue.
>
> ## 3. Throughput in training and inference.
>
> In section 5.1, we present the training throughput results for LLaMA-2 1B under two training configurations: single-GPU and a standard data-parallel (DP) setup, as summarized in Fig. 3. Using consistent hyperparameters—including batch size, sequence length, optimizer, and kernel implementation—CR-Net achieves a throughput of 13,422 tokens/s, representing a 66% improvement over the full-rank baseline (7,161 tokens/s) and a 16.6% gain over the strongest non-CR-Net baseline (CoLA, 11,506 tokens/s). Under the DP configuration, the relative performance ordering remains consistent, with CR-Net yielding the highest throughput among all compared methods. All reported values reflect end-to-end trainer-level tokens/s as measured by the training framework; no attempt was made to isolate or model communication overhead. Accordingly, the DP results should be interpreted as overall throughput under the given setup, rather than as an analysis of communication efficiency.
>
> We have also included comparisons of inference throughput for LLaMA-2 1B and training throughput for LLaMA-2 7B on a NVIDIA A100 GPU with 80GB memory. We followed the experimental setup outlined in our pre-training experiments in Appendix E.2. For the LLaMA-2 7B pre-training experiments, we utilized the maximum microbatch size that could be accommodated without triggering out-of-memory errors.
>
> Table 2: Inference throughput in LLaMA-2 1B (tokens/s).
>
> |Full-rank|LORO|CR-Net|
> |-|-|-|
> |79581|85449|116100|
>
> Table 3: Training throughput in LLaMA-2 7B (tokens/s).
>
> |Full-rank|LORO|CR-Net|
> |-|-|-|
> |1972|5990|6509|
>
>
> The result can be summarized in Table 2 and Table 3. Under the same experimental settings, CR-Net improves both training throughput and inference throughput than full-rank model. Collectively, these experiments demonstrates that CR-Net delivers consistent wall-clock time improvements over both full-rank training and recent low-rank baselines across single-GPU and multi-GPU DP configurations. We also discuss the throughput result in Section 5.1 in the revised manuscript.
>
> ## References
>
> [1] Parameter and memory efficient pretraining via low-rank Riemannian optimization
>
> ----
>
> We sincerely appreciate the reviewer’s thoughtful comments once again. We hope the responses address the concerns raised and would be happy to provide further clarification if needed.

---

> ### Author Response · Authors · 2025-11-24
>
> Thanks for the reply and the update for the score to 6. And we also appreciate the reviewer for the recommendation for CR-Net.

---

### Official Review · Reviewer_kvx9 · 2025-11-02

**Soundness:** 3
**Presentation:** 3
**Contribution:** 3
**Rating:** 6
**Confidence:** 3

**Summary:**

The authors empirically discovered that the low rank reconstruction error for the *difference* from the previous layer's activations is less than that for the layer's activations, relatively, suggesting that we should do low rank modeling for this difference instead. The scale of the previous layer's activation $Y_{l-1}^P$ going into the current layer $Y_l^P$ is a learnable parameter, balancing between whether it relies heavily on the low rank residual term or refines from the previous activation. Moreover, the first layer uses full rank.

The paper proposes activation-efficient re-computation to address the challenges of new dependencies for gradient checkpointing.

Experiments provided for pretraining Llama 2 with varying sizes up to 7B on the C4-en dataset and comparing with methods with similar low rank complexity or full rank with similar optimizer memory overhead.

**Strengths:**

- The proposed method is simple but works well. The flow of the paper is intuitive, starting from an empirical observation to developing the full method.
- The authors propose beta_0 as a scaling factor and provide a theoretical insight that under the assumption of high cosine similarity between activations, we can select a beta_0 such that the error for eq (3) is less than the regular low rank approximation, which seems reasonable to me.
- The paper includes complexity analysis

**Weaknesses:**

- The communication overhead from the residual is high in multi gpu training, although the authors argue that the computation time outweighs that. However, it is unclear if the communication volume would affect real-life large scale distributed training.

**Questions:**

- In the experiment leading to the observation, what is the value of beta_0 and how did you choose it? Do results hold with varying selections of beta_0?

---

> ### Author Response · Authors · 2025-11-22
> **Rebuttal by Authors**
>
> We are grateful for the reviewer's insightful feedback and constructive comments. We address each point in detail below.
>
> ----
>
> ## 1. Communication Volume in Real-Life Large-Scale Training
>
> We appreciate the reviewer's insightful comment. Below, we provide additional discussions and empirical details on communication volumes to address the concerns.
>
> **a. Communication in Data-Parallel Settings.**
>
> The throughput results in Figure 3 encompass CR-Net's performance in both data-parallel scenarios with multiple devices and single-GPU settings. It is evident that CR-Net consistently outperforms the baseline methods, irrespective of whether data-parallel communication overhead is included, thereby reflecting its practical efficacy in real-world data-parallel training.
>
> **b. Communication in Pipeline-Parallel Settings.**
>
> As outlined in Appendix G, our theoretical analysis for CR-Net in pipeline-parallel environments indicates that the communication overhead—in terms of time complexity and HBM memory—is negligible compared to the computation and memory savings afforded by the parameter-efficient design. To empirically validate communication efficiency, we pre-trained a LLaMA-2 70B model with CR-Net under pipeline parallelism using 8 NVIDIA A800 GPUs interconnected via NVLink, on the C4-en dataset (sequence length=256). Pipeline parallel training was implemented with the `torch.distributed.pipelining` library, with each GPU handling 10 transformer layers.
>
> The measured communication time for one forward and backward step was **1.380 seconds**, yielding a bandwidth of 192.5 GB/s. This bandwidth is approximately three times higher than the PCIe bandwidth assumed in our theoretical analysis (Table 22 in the revised manuscript). These findings demonstrate that under NVLink connectivity, **CR-Net achieves shorter communication times than theoretically projected**, further confirming that for large scale models, the communication volume does not substantially impair overall efficiency, thus supporting our theoretical conclusions.
>
> In summary, the empirical evidence reinforces that CR-Net's communication overhead is manageable in large-scale distributed training. We will update the relevant results in Appendix G to incorporate these details.
>
>
>
> ## 2. Selection for $\beta_0$
>
> In the experiment for the observation, we select the $\beta_0$ from the set $\\{0,0.2,0.4,0.6,0.8,1,2,3,5\\}$ to minimize the relative error given by (4). The optimal $\beta_0$ mostly takes the values 0.4, 0.6, and 0.8, which confirms the distribution of $\beta$ values reported in Table 20.   We have present a detail discussion in Appendix E.1 with blue highlighting.
>
> ----
>
> We sincerely appreciate the reviewer’s thoughtful comments once again. We hope the above responses fully address the concerns raised and would be happy to provide further clarification if needed.

---

> ### Author Response · Authors · 2025-11-26
> **Can we have your invaluable feedback on our rebuttals?**
>
> Dear Reviewer kvx9,
>
> We sincerely thank you for your valuable comments and appreciate the time and effort dedicated to providing constructive feedback on our submission. We have carefully considered your suggestions and made significant efforts to address them. Given the limited timeframe of the discussion period, we would greatly appreciate if you could review our rebuttal and let us know if any concerns remain. Your insights are invaluable as we strive to enhance the quality of our work.
>
> Best,
>
> The authors of paper 2550

---

### Author Response · Authors · 2025-11-22

Dear reviewers,

We have carefully considered all your comments and questions, and have addressed them one by one. The corresponding revisions have been incorporated into the main text. If you have any further questions, please do not hesitate to contact us. We sincerely appreciate your time and insightful comments.

Best,

The authors of paper 2550

---

### Author Response · Authors · 2025-12-01
**Summary of Author Responses to Reviewer Comments**

Dear area chair,

We sincerely appreciate the dedication and effort you have invested in our submission. To provide you with a comprehensive overview of the specific details during the author-reviewer discussion phase, we have outlined our responses to each of the reviewers' comments as follows.

**1. Reviewer kvx9 (Rating 6, confidence 3)**
- We present empirical results on the common communication efficiency to verify that the pipeline-parallel communication in CR-Net is minor compared to the efficiency gains from computational savings.
- We discuss how $\beta_0$ is selected in the experiments presented in Section 3.

**2. Reviewer DNpk (Rating 4, confidence 3 → Rating 6, confidence 4)**
- We discuss the advantages of CR-Net compared to the fully low-rank baseline LORO. Specifically, CR-Net achieves higher throughput and maintains stability due to its reliance solely on standard optimizers.
- We clarify how CR-Net overcomes the difficulties of low-rank pre-training in architectural design, empirical analysis, and optimization characteristics.
- We present throughput results for real-world pre-training and inference tasks with different model sizes.

**3. Reviewer BJ5z (Rating 8, confidence 4)**
- We compare CR-Net with additional baselines for low-rank pre-training in tasks involving model scaling from 60M to 1B.
- We examine the performance of CR-Net with activation compression.
- We provide additional discussion on why the full-rank first-layer is essential.
- We adjust the article structure and revise minor errors in our manuscript.

**4. Reviewer WfvT (Rating 8, confidence 5)**
- We provide an additional discussion on the results of our pre-training tasks for 13B models. We also clarify the use of re-computation in pre-training tasks, noting that re-computation is only applied to the 7B and 13B models.
- We revise minor errors in our manuscript.

Prior to the discussion period, three out of the four reviewers recommended acceptance of this paper, with ratings ranging from borderline accept to accept. Following the author rebuttal, the sole reviewer who had initially assigned a rating of 4 revised it to 6 and commented in favor of acceptance. As a result, **all four reviewers exhibited a consensus leaning towards acceptance.**

Finally, we would like to express our sincere gratitude to the Area Chair for their diligent efforts in handling our submission. We would welcome the opportunity to provide any further clarification that may be needed.

Best,

The authors of paper 2550

---

### Meta-Review · Area_Chair_G8Ud · 2025-12-18

**Summary:**

This paper introduces CR-Net, a parameter-efficient pretraining framework for LLMs that leverages the empirical observation that the inter-layer activation difference is low-rank. CR-Net reconstructs each layer’s activation from the previous layer plus a low-rank residual, modulated by a learnable interpolation/scaling factor. Reviewers find the method simple, intuitive, and practically useful, with convincing experiments at large scale and complexity analysis. Reviewers identify several concerns on distributed-training communication overhead, comparisons to low-rank weights, performance on 13B model. The rebuttal largely addresses reviewers' concerns and thus I recommend acceptance.

**Reviewer Concerns:**

The concerns on communication overhead, comparisons to low-rank weights baselines, Run-time efficiency have been addressed by the rebuttal. The concern on larger models, such as 13B remains valid and acknowledged by the authors.

**Reviewer Scores:**

One reviewer opted for borderline acceptance and two reviewers opted for acceptance. One reviewer raised score from borderline reject to borderline accept.

---

### Decision · Program_Chairs · 2026-01-26

Accept (Poster)